# When False Positive is Intolerant: End-to-End Optimization with Low FPR for Multipartite Ranking

**Peisong Wen**[1,2]    **Qianqian Xu**[1*]    **Zhiyong Yang**[2]
**Yuan He**[3]    **Qingming Huang**[1,2,4,5*]
[1] **Key Lab of Intell. Info. Process., Inst. of Comput. Tech., CAS**
[2] **School of Computer Science and Tech., University of Chinese Academy of Sciences**
[3] **Alibaba Group**
[4] **BDKM, University of Chinese Academy of Sciences**
[5] **Peng Cheng Laboratory**
{wenpeisong20z,xuqianqian}@ict.ac.cn
{yangzhiyong21,qmhuang}@ucas.ac.cn    heyuan.hy@alibaba-inc.com

## Abstract

Multipartite ranking is a basic task in machine learning, where the Area Under the receiver operating characteristics Curve (AUC) is generally applied as the evaluation metric. Despite that AUC reflects the overall performance of the model, it is inconsistent with the expected performance in some application scenarios, where only a low False Positive Rate (FPR) is meaningful. To leverage high performance under low FPRs, we consider an alternative metric for multipartite ranking evaluating the True Positive Rate (TPR) at a given FPR, denoted as TPR@FPR. Unfortunately, the key challenge of direct TPR@FPR optimization is two-fold: **a)** the original objective function is not differentiable, making gradient backpropagation impossible; **b)** the loss function could not be written as a sum of independent instance-wise terms, making mini-batch based optimization infeasible. To address these issues, we propose a novel framework on top of the deep learning framework named *Cross-Batch Approximation for Multipartite Ranking (CBA-MR)*. In face of **a)**, we propose a differentiable surrogate optimization problem where the instances having a short-time effect on FPR are rendered with different weights based on the random walk hypothesis. To tackle **b)**, we propose a fast ranking estimation method, where the full-batch loss evaluation is replaced by a delayed update scheme with the help of an embedding cache. Finally, experimental results on four real-world benchmarks are provided to demonstrate the effectiveness of the proposed method.

## 1   Introduction

The multipartite ranking is a multi-class extension of bipartite ranking, aiming to sort a dataset with multiple discrete labels in proper order. Different from general multi-class classification, the categories in multipartite ranking are arranged according to a certain attribute. In the past decade, multipartite ranking is studied in a variety of application scenarios in the field of computer vision, including age estimation[33], monocular depth estimation [12] and aesthetic visual analysis [32], *etc*.

Previous literature [13, 29, 35, 37, 38] uses Area Under the ROC Curve (AUC) as the evaluation metric for multipartite ranking. Intuitively, Hanley & McNeil [18] prove that AUC measures the probability expectation that positive instances are sorted higher than negative instances. This metric

---

*Corresponding authors.

35th Conference on Neural Information Processing Systems (NeurIPS 2021).

is less sensitive to class imbalance than other metrics like accuracy [41]. Beside AUC, other common used metrics include accuracy and mean square error (MSE) (see Sec. 2).

***However, do these metrics always reflect the model performance in different scenarios properly?***

This problem has more significance under *False Positive Rate (FPR) sensitive* application scenarios, *i.e.*, predictions leading to high false positive rates are intolerant. In these cases, the metrics mentioned above cannot properly measure the desired performance, especially when the expected FPR is low. One example of such scenarios is medical diagnosis [24, 25]. Considering diseased cases as negatives, more patients are missed if FPR is higher. Another example of churn in the telecommunications industry is provided by Mozer *et al.*[31].

Based on the above consideration, our interest is to optimize the True Positive Rate (TPR) at a fixed False Positive Rate (FPR) for multipartite ranking, denoted as TPR@FPR. For practical reasons described above, the chosen FPR is in general small. Despite the practical advantages, it is infeasible to optimize such a metric in an end-to-end manner. To be concrete, the main challenges are as follows:

(**C1**) TPR@FPR is a constrained combinatorial optimization problem, where the gradients of the objective function are not directly available even when proper surrogate losses are utilized.

(**C2**) The loss function could not be expressed as a sum of independent instance-wise terms, making the stochastic optimization unavailable.

To solve these problems, we propose a novel method named *Cross-Batch Approximation for Multipartite Ranking (CBA-MR)*. In a nutshell, the main contributions of this paper are summarized as follows:

- We consider a new evaluation metric TPR@FPR for multipartite ranking in FPR sensitive scenarios. This metric focuses on model performance with a low FPR, which is consistent with practical requirements.

- To optimize the TPR@FPR metric, we propose Cross-Batch Approximation for Multipartite Ranking. The main idea is to relax the tricky hard constraint on FPR as a probability-based loss function. Specifically, we propose a random-walk model to measure the possibility that a negative instance will affect short-term FPR value. On top of this model, we propose a differentiable loss function that automatically focuses on hard negative instances.

- Motivated by the slow feature drift phenomenon, we introduce a nonparametric cross-batch cache module to approximate the ranking of negative examples rapidly. An upper bound of the approximation error is provided to exhibit the feasibility of approximation.

## 2 Related Work

**Multipartite Ranking & Ordinal Regression.** Existing methods on ordinal regression can be roughly divided into two technical routes [16]: classification-based approaches and threshold-based approaches. Classification-based approaches transform the ordinal regression problem into the general classification problem. The most common seen transformation is F&H [11], which decomposes the $M$ classes original regression into $M-1$ binary problems, *i.e.*, for each $k \in \{1, \cdots, M-1\}$, whether the rank of the category of an instance is larger than $k$? This method has been applied in age estimation[33] and depth estimation [12]. Lin and Li [23] define an extended binary example $\boldsymbol{x}^{(k)} = (\boldsymbol{x}, k), y^{(k)} = 2I[y < k] - 1$ for each example $(\boldsymbol{x}, y)$ and sub-problem $k \in \{1, \cdots, M-1\}$, and learns a single binary classifier on the extended data. Most of the above methods use manually extracted features and classifiers based on support vector machine (SVM) [2], while the neural network is first adopted by Cheng *et al.*[4], and further developed in [9, 27, 1].

The basic idea of threshold-based approaches is to sort the samples firstly, and then determine the categories of samples by thresholds. Multipartite ranking and threshold-based ordinal regression are essentially similar, since the purpose of both problems is to learn scores for instances indicating the ranking. The main difference between these two problems is that multipartite ranking only requires to sort the queries, while ordinal regression needs to further seek discrete labels. Uematsu & Lee [38] prove that some solutions to ordinal regression [22, 6, 30] are special cases of multipartite ranking. Along this technical route, Chu and Ghahramani [5] assume that the latent function from the input space to the output space is a Gaussian process, and solve the optimal parameters through a multilayer

perceptron (MLP). Liu *et al.*[28] further extend a non-conjugate likelihood to estimate parameters of the Gaussian process in a stochastic mini-batch manner. Quevedo *et al.*[35] further develop F&H[11], exploring another decomposition based on decision directed acyclic graph. Other early literature modifies SVM or ensemble models from classification problems. Chu and Keerthi propose SVOR [6], which learns a linear transformation, and classifies the samples into corresponding categories according to the distance to the discriminant plane. Lin and Li *et al.*[22] propose an ensemble model structure based on large-margin bounds. Deep learning is also applied along this technical route [26].

Although various methods have been proposed for ordinal regression or multipartite ranking, these methods still leave the FPR sensitive scenarios remaining, since they focus on metrics like AUC, accuracy or MSE. To fill this gap, we consider a new metric TPR@FPR that focuses on low FPRs, and propose a novel framework to optimize it in an end-to-end manner.

**Constrained Optimization on Classification.** Optimizing the TPR@FPR metric can be viewed as a constrained optimization problem. Existing literature provides solutions to some constrained optimization problems. For example, to maximize precision at the top of a ranked list (P@K), Kar *et al.*[21] propose surrogate functions for this problem. However, this work fails when constraints depend on the ground truth labels. To solve this kind of constraints in a special case, *i.e.*, linear models, Goh *et al.*[15] propose to use ramp penalty to accurately quantify costs. For non-linear models, Eban *et al.*[10] introduce Lagrange multiplier to transform the problem into a min-max problem, and solve it with iterative SGD. Mackey *et al.*[29] use kernel quantile estimator to approximate the constraint. At first glance, TPR@FPR can be solved similarly to these two methods, but the convergence cannot be guaranteed when the model is non-linear or even non-convex, and the mini-batch-based Lagrange multiplier has a large update variance. Moreover, the method of Mackey *et al.*[29] will lead to training collapse when the excepted FPR is low. Recent work [19, 20] proposes to apply deep neural networks to learn a surrogate function for black-box metrics. However, this route also requires gradients of the metric.

In short, these existing methods cannot solve our problem through simple modifications. In this work, to solve TPR@FPR optimization problem, we propose an end-to-end framework based on cross-batch approximation.

# 3 Methodology

In this section, we will present the details of our proposed framework. In Sec. 3.1, we provide a formal definition of the multipartite ranking problem and the TPR@FPR metric. In Sec. 3.2, we present an overview of our proposed method. Finally, in Sec. 3.3 and Sec. 3.4, we describe the details of the cross-batch ranking estimation and the cross-batch cache module, respectively.

## 3.1 Problem Settings and Metrics

**Notations.** Consider a dataset with $N$ instances and $M$ categories: $\mathcal{D} = \{(\boldsymbol{x}_i, y_i) | \boldsymbol{x}_i \in \mathcal{X}, y_i \in \mathcal{Y}\}_{i=1}^N$, where $\mathcal{X}$ is the input image space and $\mathcal{Y} = \{r_1, r_2, \cdots, r_M\}$ is the corresponding category space. In multipartite ranking, the categories in $\mathcal{Y}$ satisfy an ordered ranking $r_1 \prec r_2 \prec \cdots \prec r_M$. Generally, the objective is to learn a score function $f : \mathcal{X} \to \mathbb{R}$, such that the instances with label $r_i$ are assigned higher scores than those with label $r_j$, whenever $r_i \succ r_j$.

**Problem Decomposition.** As explained in Sec. 1, we decompose the multipartite ranking problem into $M-1$ bipartite ranking sub-problems. The $k$-th sub-problem is to perform a bipartite ranking with $\mathcal{D}_k^+ = \{(\boldsymbol{x}_i, y_i) \in \mathcal{D} | y_i \succ r_k\}$ being the set of positive instance, and $\mathcal{D}_k^- = \{(\boldsymbol{x}_i, y_i) \in \mathcal{D} | y_i \preceq r_k\}$ being that of the negative instance. For the sake of convenience, we denote the $i$-th instance from $\mathcal{D}_k^+$ and its corresponding score as $\boldsymbol{x}_i^{k+}$ and $s_i^{k+}$ respectively, and denote $\boldsymbol{x}_i^{k-}, s_i^{k-}$ as those for the instances from $\mathcal{D}_k^-$.

Given such a decomposition, we then define the TPR@FPR metric based on the performance of the bipartite sub-problems:

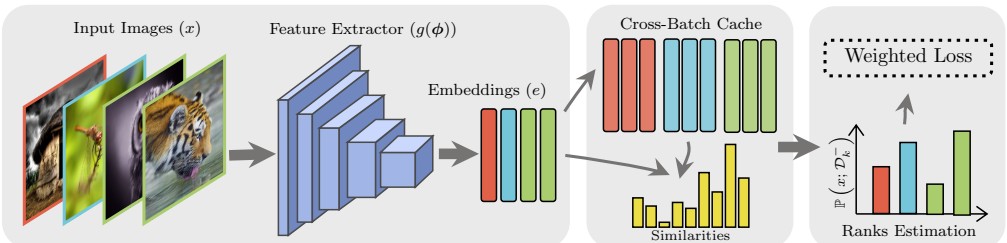

Figure 1: An overview of the proposed method. After extracting embeddings for input images, we update the cache block, and estimate the rankings of instances in caches. Then, the rankings of instances in the current mini-batch are estimated with similarities in caches. Finally, the rankings are transformed into probabilities for weighting the loss function.

**Definition 1.** *Given a bipartite ranking dataset with a positive subset $\mathcal{D}^+ = \{\boldsymbol{x}_i^+\}_i$ and a negative subset $\mathcal{D}^- = \{\boldsymbol{x}_i^-\}_i$, the TPR at a fixed FPR $\alpha$ (TPR@FPR) for a scoring function $f_{\boldsymbol{\theta}}$ is defined as*

$$TPR@FPR(\mathcal{D}^+, \mathcal{D}^-, f_{\boldsymbol{\theta}}, \tau(\alpha)) = \frac{1}{|\mathcal{D}^+|} \sum_{i=1}^{|\mathcal{D}^+|} I[s_i^+ > \tau(\alpha)], \tag{1}$$

*where the indicator function $I[x]$ equals to 1 if $x$ is true and 0 otherwise. $s_i^+, s_i^-$ refer to $f_{\boldsymbol{\theta}}(\boldsymbol{x}_i^+), f_{\boldsymbol{\theta}}(\boldsymbol{x}_i^-)$, respectively. $\tau(\alpha)$ is the minimum classification threshold ensuring that the $FPR$ is smaller than $\alpha$. Mathematically, this could be expressed as:*

$$\tau(\alpha) = \min_{\hat{\tau}} \hat{\tau} \quad s.t. \quad \frac{1}{|\mathcal{D}^-|} \sum_{i=1}^{|\mathcal{D}^-|} I[s_i^- > \hat{\tau}] \leq \alpha. \tag{2}$$

Based on our decomposition scheme, TPR@FPR for multipartite ranking can be naturally expressed as a weighted average of TPR@FPR for bipartite ranking defined as Def. 2.

**Definition 2.** *Given a multipartite ranking dataset as defined above, the True Positive Rate at a fixed False Positive Rate (TPR@FPR) for a scoring function $f_{\boldsymbol{\theta}}$ is defined as*

$$TPR@FPR(\mathcal{D}, f_{\boldsymbol{\theta}}, \alpha) = \frac{1}{\sum_{k=1}^{M-1} \lambda_k} \sum_{k=1}^{M-1} \lambda_k TPR@FPR(\mathcal{D}_k^+, \mathcal{D}_k^-, f_{\boldsymbol{\theta}}, \tau_k(\alpha)), \tag{3}$$

*where $\lambda_k$ is the weight of the $k$-th sub-problem.*

Given the definitions above, our goal is to maximize the TPR@FPR($\mathcal{D}, f_{\boldsymbol{\theta}}, \alpha$). It is abbreviated as TPR@FPR in the rest of this paper if it does not lead to ambiguity.

As mentioned in Sec. 1, optimizing such a complicated metric is non-trivial especially in the end-to-end training regime. In the rest of this section, we will propose an efficient method to optimize the metric approximately.

### 3.2 Overview

An overview of our proposed method is shown in Fig. 1. In this paper, we implement the scoring function $f_{\boldsymbol{\theta}}$ as a deep neural network. Specifically, we first extract an embedding $\boldsymbol{e} \in \mathbb{R}^d$ from an input image $\boldsymbol{x}$ with a deep model $g_{\boldsymbol{\phi}}$: $\boldsymbol{e} = g_{\boldsymbol{\phi}}(\boldsymbol{x})$, where $\boldsymbol{\phi}$ is the set of parameters. Afterward, we learn the ranking score $s \in \mathbb{R}$ with a linear transformation $\boldsymbol{w} \in \mathbb{R}^d$: $s = \boldsymbol{w}^T \boldsymbol{e}$. Denote $\boldsymbol{\theta} = \{\boldsymbol{\phi}, \boldsymbol{w}\}$ and $f_{\boldsymbol{\theta}}(\cdot) = \boldsymbol{w}^T g_{\boldsymbol{\phi}}(\cdot)$. Then our goal is to learn $\boldsymbol{\theta}$ that approximately maximizes TPR@FPR.

Recall the main challenges summarized in Sec. 1. To address **(C1)**, we propose a cross-batch ranking estimation module to make the gradient backpropagation feasible. Meanwhile, to address **(C2)**, we propose a cross-batch caching module to approximate the full-batch update in a few iterations.

### 3.3 Cross-Batch Ranking Estimation

Firstly, we consider a reformulation of the objective to remove the constraints. Here, we denote $s_{(i)}^{k-}$ as the $i$-th largest score of the negative instances in the observed dataset $\mathcal{D}_k^-$, such that

$$s_{(1)}^{k-} \geq s_{(2)}^{k-} \geq \cdots s_{(|\mathcal{D}_k^-|)}^{k-}.$$

Obviously $\tau_k(\alpha) = s_{(\lceil \alpha|\mathcal{D}_k^-|\rceil)}^{k-}$, and thus the objective function could then be reformulated as an unconstrained problem:

$$\min_{\boldsymbol{\theta}} \quad \mathcal{L}_{0,1} = \frac{1}{\sum_{k=1}^{M-1} \lambda_k} \sum_{k=1}^{M-1} \frac{\lambda_k}{|\mathcal{D}_k^+|} \sum_{i=1}^{|\mathcal{D}_k^+|} \ell_{0,1}(s_i^{k+} - s_{(\lceil \alpha|\mathcal{D}_k^-|\rceil)}^{k-}), \tag{4}$$

where $\ell_{0,1}(x)$ equals 1 if $x$ is negative and 0 otherwise, and $\lceil \cdot \rceil$ is the ceiling function. Nonetheless, it is easy to see that the 0-1 loss $\ell_{0,1}$ here is not differentiable, and thus directly optimizing the original problem is NP-hard. To handle this issue, the 0-1 loss could be replaced by a differentiable surrogate function $\ell$ such as logistic loss, exponential loss or hinge loss [14]. Among them, we simply implement $\ell$ with logistic loss, *i.e.*, $\ell(x) = \log(1 + \exp(-x))$, which leads to the following objective:

$$\min_{\boldsymbol{\theta}} \quad \mathcal{L} = \frac{1}{\sum_{k=1}^{M-1} \lambda_k} \sum_{k=1}^{M-1} \frac{\lambda_k}{|\mathcal{D}_k^+|} \sum_{i=1}^{|\mathcal{D}_k^+|} \ell(s_i^{k+} - s_{(\lceil \alpha|\mathcal{D}_k^-|\rceil)}^{k-}). \tag{5}$$

Unfortunately, even with a proper surrogate loss, there are still two main challenges to optimize the objective: **1)** optimizing the order statistic $s_{(\lceil \alpha|\mathcal{D}_k^-|\rceil)}^{k-}$ is non-trivial since the gradient backpropagation is impossible without a full-batch scan; and **2)** $s_{(\lceil \alpha|\mathcal{D}_k^-|\rceil)}^{k-}$ is the only negative instance that explicitly appears in the loss function, making information from other negative instances totally ignored.

To address this issue, we turn to seek a proper approximated objective function where all available negative instances are included. Note the objective function in Eq. (5) essentially tries to punish hard negative instances. In this sense, we expect the approximated objective function to focus on the top $\lceil \alpha|\mathcal{D}_k^-|\rceil$ hard negatives instead of merely $s_{(\lceil \alpha|\mathcal{D}_k^-|\rceil)}^{k-}$. To do this, we introduce a set of probabilistic weights to attend to the hard negative instances. Since the model parameters are constantly changing as the training process goes on, the score ranking also enjoys a dynamic nature. Hence, Our main idea is to consider all the negative examples that may reach the top $\lceil \alpha|\mathcal{D}_k^-|\rceil$ ranking list within a short period of training iterations as the potential candidates of the hard negatives. Formally, given the negative sample set $\mathcal{A}$, we denote the probability that the score of a negative example $\boldsymbol{x}$ reaches the top $\lceil \alpha|\mathcal{A}|\rceil$ ranking list within a time window as $\mathbb{P}(\boldsymbol{x}; \mathcal{A})$. Then come to a dynamic approximated problem as:

$$\min_{\boldsymbol{\theta}} \quad \mathcal{L} = \frac{1}{\sum_{k=1}^{M-1} \lambda_k} \sum_{k=1}^{M-1} \frac{\lambda_k}{|\mathcal{D}_k^+||\mathcal{D}_k^-|} \sum_{i=1}^{|\mathcal{D}_k^+|} \sum_{j=1}^{|\mathcal{D}_k^-|} \ell(s_i^{k+} - s_j^{k-}) \mathbb{1}[\mathrm{rk}(\boldsymbol{x}_j^{k-}) = \lceil \alpha|\mathcal{D}_k^-|\rceil]. \tag{6}$$

$$\min_{\boldsymbol{\theta}} \quad \mathcal{L} = \frac{1}{\sum_{k=1}^{M-1} \lambda_k} \sum_{k=1}^{M-1} \frac{\lambda_k}{|\mathcal{D}_k^+||\mathcal{D}_k^-|} \sum_{i=1}^{|\mathcal{D}_k^+|} \sum_{j=1}^{|\mathcal{D}_k^-|} \ell(s_i^{k+} - s_j^{k-}) \mathbb{P}(\boldsymbol{x}_j^{k-}; \mathcal{D}_k^-). \tag{7}$$

Here we reweight different loss terms by $\mathbb{P}(\boldsymbol{x}_j^{k-}; \mathcal{D}_k^-)$ to focus on the potential hard negatives.

The problem then becomes **how to estimate** $\mathbb{P}(\boldsymbol{x}_j^{k-}; \mathcal{D}_k^-)$. To discuss the approximation guarantee from Eq. (5) to Eq. (7), we provide a sufficient condition under which the objectives are consistent with the performance increasing. Here we consider a sub-problem and ignore the subscript $k$. Denote $\hat{\mathcal{R}}_p^\ell$ as the empirical risk of TPR@FPR (as defined in Eq. (5)), and $\hat{\mathcal{R}}_p^\ell$ as the empirical risk of the approximated version (as defined in Eq. (7)), i.e.,

$$\hat{\mathcal{R}}^\ell = \frac{1}{|\mathcal{D}^+|} \sum_{\boldsymbol{x}_i^+ \in \mathcal{D}^+} \ell(s_i^+ - s_j^-), \tag{8}$$

$$\hat{\mathcal{R}}_p^\ell = \frac{1}{|\mathcal{D}^+|} \sum_{\boldsymbol{x}_i^+ \in \mathcal{D}^+, \boldsymbol{x}_j^- \in \mathcal{D}^-} \ell(s_i^+ - s_j^-)\mathbb{P}(s_j^-; \mathcal{D}^-), \tag{9}$$

we provide a sufficient condition for $\hat{\mathcal{R}}_p^\ell \to \hat{\mathcal{R}}^\ell$ in the following theorem (see Appendix A.1 for the proof and the simulation experiment).

**Theorem 1.** *Given a probability estimation $\mathbb{P}(\cdot; \mathcal{D}^-) : \mathbb{R} \mapsto (0, 1)$, and a surrogate loss function $\ell(\cdot)$, denote $\ell_{ij} = \ell(s_i^+ - s_j^-), p_j = \mathbb{P}(\boldsymbol{x}_j^-; \mathcal{D}^-), \beta = \lceil \alpha|\mathcal{D}^-| \rceil, \mathcal{D}_\beta^- = \mathcal{D}^-/\{\boldsymbol{x}_{(\beta)}^-\}$, and $\mathcal{I}_\beta = \mathcal{D}^+ \times \mathcal{D}_\beta^-$; denote $\hat{\mathbb{E}}_{\boldsymbol{x}^+ \in \mathcal{D}^+}[z]$ as the empirical expectation of $z$ over $\mathcal{D}^+$, and similarly for $\hat{\mathbb{E}}_{\boldsymbol{x}^- \in \mathcal{D}_\beta^-}[z], \hat{\mathbb{E}}_{\boldsymbol{x}^+, \boldsymbol{x}^- \in \mathcal{I}_\beta}[z]$; denote $\sigma = Var_{\boldsymbol{x}_i^+ \in \mathcal{D}^+, \boldsymbol{x}_i^- \in \mathcal{D}^-}[s_i^+ - s_i^-], \delta_{ij} = (s_i^+ - s_i^-)/\sigma, \bar{\delta}_i = (s_i^+ - \sum_{\boldsymbol{x}_j \in \mathcal{D}^-} s_j^- p_j)/\sigma$. Assume $\sum_{\boldsymbol{x}_j \in \mathcal{D}^-} p_j = 1$.*

*We have $\hat{\mathcal{R}}_p^\ell \to \hat{\mathcal{R}}^\ell$ when $\sigma \to 0$ if the following conditions are met:*

    *(a) $\ell$ is convex, monotonically decreasing, $\ell(0) > 0$, and $(\ell')^2 - \ell'' \cdot \ell \geq 0$;*

    *(b) $\sum_{\boldsymbol{x}_j \in \mathcal{D}^-} s_j^- p_j \leq s_\beta^-$;*

    *(c) $\inf_{u \in (0,1), v = 1-u}[A_u - B_v] \leq 0$, where*

$$A_u = (|\mathcal{D}^-| - 1)(\hat{\mathbb{E}}_{\boldsymbol{x}^+, \boldsymbol{x}^- \in \mathcal{I}_\beta}[\ell^{1/u}])^u, \tag{10}$$

$$B_v = (1 - p_\beta)\hat{\mathbb{E}}_{\boldsymbol{x}^+ \in \mathcal{D}^+}[\ell_{i\beta}]/(\hat{\mathbb{E}}_{\boldsymbol{x}^- \in \mathcal{D}_\beta^-}[p^{1/v}])^v. \tag{11}$$

Based on Thm. 1, we provide an instantiation of $\mathbb{P}(\boldsymbol{x}_j^{k-}; \mathcal{D}_k^-)$ in the reset of this subsection. Our key idea here is to measure such possibilities with the randomness of the relative ranking changes. Moreover, by a hypothesis test results listed in Appendix B, we observe that short-term ranking changes will tend to stabilize at white noises as the model training becomes stable with statistical significance. Next, we will propose a basic assumption to capture such a phenomenon. Concretely, denote a subset of the dataset as $\mathcal{A} \subseteq \mathcal{D}$, and the descending rank of an example $\boldsymbol{x} \in \mathcal{A}$ under parameters $\boldsymbol{\theta}$ as $R(\boldsymbol{x}; \mathcal{A}, \boldsymbol{\theta}) = \sum_{\boldsymbol{x}_i \in \mathcal{A}} I[f_{\boldsymbol{\theta}}(\boldsymbol{x}) < f_{\boldsymbol{\theta}}(\boldsymbol{x}_i)]$, the assumption is described as follows:

**Assumption 1.** *Denote the parameters at timestamp $t$ as $\boldsymbol{\theta}_t$. In a relatively short period of time $T$, the score ranking on a certain set $\mathcal{A}$ obeys a random walk. Equivalently, the difference sequence $\{R(\boldsymbol{x}; \mathcal{A}, \boldsymbol{\theta}_{t+1}) - R(\boldsymbol{x}; \mathcal{A}, \boldsymbol{\theta}_t)\}_t$ is a white noise sequence, i.e., its expectation is 0, and for any $k \geq 2$, the $k$-order correlation with $t$ is 0.*

Based on the above assumption, the probability $\mathbb{P}(\boldsymbol{x}; \mathcal{A})$ can be viewed as the probability of random walk from $R(\boldsymbol{x}; \mathcal{A}, \boldsymbol{\theta})$ to $\lceil \alpha|\mathcal{A}| \rceil$, or vice versa from $\lceil \alpha|\mathcal{A}| \rceil$ to $R(\boldsymbol{x}; \mathcal{A}, \boldsymbol{\theta})$. When $\boldsymbol{\theta}$ tends to convergence, it can be seen that only three events occur each time: the rank of a specific instance decreases by 1, increases by 1, or remains the same. In order to facilitate the analysis, we consider the following continuous form of the assumption.

**Assumption 2.** *In a relatively short period of time $T$, the normalized score ranking $R(\boldsymbol{x}; \mathcal{A}, \boldsymbol{\theta})/|\mathcal{A}|$ of a specific instance $\boldsymbol{x}$ on a certain set $\mathcal{A}$ takes a step per $\Delta t$ time, where three conditions are possible: 1) $R(\boldsymbol{x}; \mathcal{A}, \boldsymbol{\theta}_{t+\Delta t})/|\mathcal{A}| = R(\boldsymbol{x}; \mathcal{A}, \boldsymbol{\theta}_t)/|\mathcal{A}| + \Delta s$ with probability $p/2$; 2) $R(\boldsymbol{x}; \mathcal{A}, \boldsymbol{\theta}_{t+\Delta t})/|\mathcal{A}| = R(\boldsymbol{x}; \mathcal{A}, \boldsymbol{\theta}_t)/|\mathcal{A}| - \Delta s$ with probability $p/2$; or 3) $R(\boldsymbol{x}; \mathcal{A}, \boldsymbol{\theta}_{t+\Delta t})/|\mathcal{A}| = R(\boldsymbol{x}; \mathcal{A}, \boldsymbol{\theta}_t)/|\mathcal{A}|$ with probability $1 - p$. Here, $0 < p < 1$.*

Finally, based on Asmp. 2, the following proposition shows that $\mathbb{P}(\boldsymbol{x}; \mathcal{A})$ enjoys a simple expression (see Appendix A.1 for proof):

**Proposition 1.** *Assume $\Delta t \to 0$, $\Delta s \to 0$, and $\Delta s^2 = \beta \Delta t$, the probability that the ranking of $\boldsymbol{x}$ reaches the rank $\lceil \alpha|\mathcal{A}| \rceil$ after a period of time $T$ is as follows:*

$$\mathbb{P}(\boldsymbol{x}; \mathcal{A}) \approx \frac{1}{\sqrt{2\pi}\sigma} \exp\left(-\frac{(R(\boldsymbol{x}; \mathcal{A}, \boldsymbol{\theta})/|\mathcal{A}| - \alpha)^2}{2\sigma^2}\right), \tag{12}$$

where $\sigma = \sqrt{p\beta T/2}$.

According to Prop. 1, given the current ranking of negative examples, the probability that a negative instance reaches the top $\alpha|\mathcal{D}_k^-|$ ranking list in a few iterations can be approximated with a Gaussian process. Moreover, for $\mathbb{P}(\boldsymbol{x}; \mathcal{A})$, the only dependence on time $T$ is from $\sigma$, a tunable hyperparameter. In this way, one can perform gradient backpropagation on top of the proposed loss function, as long as a proper estimation of rank $R(\boldsymbol{x}; \mathcal{A}, \boldsymbol{\theta})$ is available.

## 3.4 Cross-Batch Cache

Despite that $\mathbb{P}(\boldsymbol{x}_j^{k-}; \mathcal{D}_k^-)$ in Eq. (7) can be approximated with Eq. (12), the loss function is still not a sum of independent instance-wise terms, since the ranks are not independent of other instances. Moreover, re-calculating the ranking under the current parameter $\boldsymbol{\theta}$ after each update will significantly slow down the training process, which motivates us to propose a rapid estimation for the score ranks.

The main computational bottleneck is the frequent update of the full-batch feature embeddings from input images, where historical embeddings are usually considered as out-of-date once the update is finished. However, recent literature [3, 39] on deep metric learning shows that the embeddings generated in the past iterations are still valuable. Wang *et al.*[39] observe an interesting phenomenon called *slow drift*, *i.e.*, the embedding drift of the same sample within a few steps will decrease as the training is stable, thus the past embeddings can approximate the embeddings under the current model parameters.

Inspired by this phenomenon, we adopt a caching mechanism to perform a delayed embedding update. Concretely, for each category $k \in \{1, \cdots, M\}$, we set up a cache $\mathcal{M}_k = \{\tilde{\boldsymbol{e}}_i = g_{\tilde{\boldsymbol{\phi}}}(\boldsymbol{x}_i)|(\boldsymbol{x}_i, y_i) \in \mathcal{D}, y_i = k\}$, and $\mathcal{M}_k^- = \cup_{i=1}^k \mathcal{M}_i$, where $\tilde{\boldsymbol{\phi}}$ is the model parameters in past iterations. To balance the per-category sample size, the size of the cache for each category is limited to $N_c$. At each iteration, the oldest embeddings from $\mathcal{M}_k$ are popped out from the cache, and the newly generated embeddings are pushed into the corresponding cache.

Given a mini-batch $\mathcal{B} \subset \mathcal{D}$, we generate embeddings for instances in $\mathcal{B}$, and then update the caches as described above. Afterward, the ranks of instance scores under the current parameters $\boldsymbol{\theta}$ are estimated with the cache. Specifically, $\hat{R}(\boldsymbol{x}_i^{k-}; \mathcal{D}_k^-, \boldsymbol{\theta})$, the rank of $\boldsymbol{x}_i^{k-} \in \mathcal{D}_k^-$, is approximated by the ranks of embeddings in the corresponding cache $\mathcal{M}_k^-$ according to the feature similarities $a_j$:

$$\hat{R}(\boldsymbol{x}_i^{k-}; \mathcal{D}_k^-, \boldsymbol{\theta}) = \sum_{\tilde{\boldsymbol{e}}_j \in \mathcal{M}_k^-} a_j \hat{R}(\boldsymbol{x}_j; \mathcal{M}_k^-, \boldsymbol{\theta}), \tag{13}$$

$$a_j = \exp(-\gamma\langle\boldsymbol{e}_i^{k-}, \tilde{\boldsymbol{e}}_j\rangle)/\sum_{\tilde{\boldsymbol{e}}_l \in \mathcal{M}_k^-} \exp(-\gamma\langle\boldsymbol{e}_i^{k-}, \tilde{\boldsymbol{e}}_l\rangle), \tag{14}$$

where $\gamma$ is a hyperparameter, $\langle\cdot, \cdot\rangle$ refers to inner product, and $\hat{R}(\boldsymbol{x}_j; \mathcal{M}_k^-, \boldsymbol{\theta})$ is a local estimation of $R(\boldsymbol{x}_j; \mathcal{D}_k^-, \boldsymbol{\theta})$ over the cache list $\mathcal{M}_k^-$. Afterward, $\mathbb{P}(\boldsymbol{x}_j^k; \mathcal{D}_k^-)$ is obtained with Eq. (12):

$$\mathbb{P}(\boldsymbol{x}_j^k; \mathcal{D}_k^-) = \frac{1}{\sqrt{2\pi}\sigma} \exp\left(-\frac{(\hat{R}(\boldsymbol{x}_j^k; \mathcal{D}_k^-, \boldsymbol{\theta})/|\mathcal{M}_k^-| - \alpha)^2}{2\sigma^2}\right), \tag{15}$$

where $\sigma$ is a tunable hyperparameter.

Moreover, we find that using Eq. (15) alone will lead to significant fluctuations, since the magnitude of $\mathbb{P}(\boldsymbol{x}_i^{k-}; \mathcal{D}_k^-)$ might vary greatly across different mini-batches. Therefore, we turn to use the normalized form to unify the magnitude of $\mathbb{P}(\boldsymbol{x}_i^{k-}; \mathcal{D}_k^-)$:

$$\tilde{\mathbb{P}}(\boldsymbol{x}_i^{k-}; \mathcal{D}_k^-) = \mathbb{P}(\boldsymbol{x}_i^{k-}; \mathcal{D}_k^-)/\sum_{\boldsymbol{x}_j \in \mathcal{D}_k^- \cap \mathcal{B}} \mathbb{P}(\boldsymbol{x}_j; \mathcal{D}_k^-). \tag{16}$$

To ensure the approximation is acceptable, we have an upper bound for the error of ranking estimation in Prop. 2. (See Appendix A.2 for proof):

**Proposition 2.** *Assume that $g_{\boldsymbol{\phi}}$ is L-Lipschitz continuous on $\boldsymbol{\phi}$, and $\|\boldsymbol{\phi} - \tilde{\boldsymbol{\phi}}\|_2 \leq \epsilon$, the error of ranking estimation on $\mathcal{A}$ is*

$$|R(\boldsymbol{x}; \mathcal{A}, \boldsymbol{\theta}) - R(\boldsymbol{x}; \mathcal{A}, \tilde{\boldsymbol{\theta}})| \leq \sum_{i \in \mathcal{A}} I[2\epsilon L > |\boldsymbol{w}^T(g_{\boldsymbol{\phi}}(\boldsymbol{x}) - g_{\boldsymbol{\phi}}(\boldsymbol{x}_i))|/\|\boldsymbol{w}\|_2] \tag{17}$$

*where $\tilde{\boldsymbol{\theta}} = \{\tilde{\boldsymbol{\phi}}, \boldsymbol{w}\}$.*

Table 1: Statistics of three datasets.

| Dataset | Training samples | Test / Val samples | Categories | Application |
|---------|------------------|--------------------|------------|-------------|
| DR | 24,587 | 5,271 | 5 | medical diagnosis |
| UTKFace | 16,591 | 3,556 | 7 | age estimation |
| NSFW | 38,822 | 8,321 | 3 | content review |
| AgePrediction | 152,295 | 32,639 | 7 | age estimation |

Prop. 2 shows that $R(\boldsymbol{x}; \mathcal{A}, \tilde{\boldsymbol{\theta}})$ provides a reasonable approximation of $R(\boldsymbol{x}; \mathcal{A}, \boldsymbol{\theta})$, if the predictions of instances are sufficiently scattered. Therefore, we can estimate the ranks in $\mathcal{M}_k^-$ by $R(\boldsymbol{x}; \mathcal{M}_k^-, \tilde{\boldsymbol{\theta}})$, which can be obtained at an ignorable cost of $O(MN_c \times (d + \log(MN_c)))$ time complexity.

Together with cross-batch ranking estimation, we decompose the objective into instance independent items and optimize it in a mini-batch manner. In this way, the two main challenges mentioned in the introduction are solved, and the TPR@FPR metric can be approximately optimized by mini-batch gradient descent. We present a summary of the detailed process in Appendix C.

## 4  Experiments

### 4.1  Datasets

The experiments are conducted on four benchmarks, where the downstream applications include medical diagnosis, age estimation and content review. Each dataset is split into training set, validation set and test set at a ratio of $0.7 : 0.15 : 0.15$. The statistics of these datasets are provided in Tab. 1.

- **Diabetic Retinopathy Detection (DR)** [2]**.** Diabetic retinopathy is a complication of diabetes, which is a main cause of blindness in the working-age population. The Diabetic Retinopathy Detection dataset contains 35,126 optical images of the retinas obtained from patients, each image is labeled as one of 5 levels: 0-No DR, 1-Mild DR, 2-Moderate DR, 3-Severe DR and 4-Proliferative DR.

- **UTKFace** [3]**.** The UTKFace [42] dataset is a popular benchmark for ordinal regression. This dataset contains 23,707 face images with ages ranging from 0 to 116 years old. Following previous work [8], we divide the labels into 7 ordinal groups according to ages: baby (0-3), child (4-12), teenager (13-19), young adult (20-30), adult (31-45), middle-aged (46-60) and senior (elder than 61). We use the aligned and cropped faces provided officially.

- **NSFW** [4]**.** The NSFW dataset is a public available dataset for *Not Suitable For Work* images detection. After filtering the broken links, we select the following three types of images from this dataset: 0-neutral, 1-sexy, 2-porn, which have a total of 55,459 images.

- **AgePrediction** [5]**.** The AgePrediction dataset is a large scale facial age prediction dataset. This dataset contains 217k face images. The age groups are split as in UTKFace dataset.

### 4.2  Implementation details

**Network architecture.** The feature extractor is implemented with ReXNet200 [17], the state-of-the-art CNN on image classification tasks, which is initialized with the pretrained model on ImageNet [36]. The backbone takes an image of size $224 \times 224 \times 3$ as input, which is normalized by subtracting the mean and dividing the standard deviation on ImageNet, and outputs a 2560-d embedding.

**Optimization strategy.** In the training stage, we apply data augmentation for data preprocessing, in which the augmentation policies are searched in ImageNet with auto-augmentation [7]. The objective function is optimized with the stochastic gradient descent (SGD) optimizer with a momentum 0.9, and the batch size is set to 64. The learning rate is initialized as 0.01, and decays by 0.1 per 30

---

[2]https://www.kaggle.com/c/diabetic-retinopathy-detection. Licensed MIT.

[3]https://susanqq.github.io/UTKFace. Licensed Data files © Original Authors.

[4]https://github.com/alex000kim/nsfw_data_scraper. Licensed MIT.

[5]https://www.kaggle.com/mariafrenti/age-prediction.

Table 2: Quantitative results of our proposed method and the competitors. TPR@FPR (%) and AUC (%) are reported. The best and the second best results are highlighted in soft red and soft blue, respectively.

| Datasets | DR | | | | UTKFace | | | | NSFW | | | | AgePrediction | | |
|---|---|---|---|---|---|---|---|---|---|---|---|---|---|---|---|
| Metrics | TPR@FPR | | | AUC | TPR@FPR | | | AUC | TPR@FPR | | | AUC | TPR@FPR | | |
| | 0.003 | 0.01 | 0.03 | | 0.003 | 0.01 | 0.03 | | 0.003 | 0.01 | 0.03 | | 0.003 | 0.01 | 0.03 |
| CE | 35.2 | 47.4 | 61.0 | 84.4 | 33.1 | 55.8 | 80.9 | 94.6 | 53.6 | 71.8 | 96.5 | 96.9 | 11.3 | 38.6 | 53.1 |
| NNRank [4] | 32.8 | 45.3 | 61.3 | 85.4 | 35.7 | 59.7 | 75.7 | 94.5 | 54.8 | 71.3 | 96.7 | 97.5 | – | – | – |
| F&H [11] | 36.0 | 48.4 | 63.5 | 85.8 | 37.5 | 63.0 | 80.9 | 95.7 | 52.2 | 71.9 | 96.3 | 97.4 | 11.0 | 36.4 | 48.9 |
| CR-DDAG [35] | 21.2 | 41.3 | 53.4 | 84.2 | 28.7 | 57.8 | 77.1 | 93.7 | 23.3 | 63.4 | 95.4 | 96.1 | – | – | – |
| PR-DDAG [35] | 25.3 | 45.7 | 60.3 | 85.0 | 24.9 | 60.1 | 82.5 | 96.0 | 49.6 | 71.0 | 96.6 | 97.1 | – | – | – |
| SoftLabel [9] | 36.8 | 48.2 | 64.3 | 85.9 | 39.5 | 63.2 | 82.0 | 96.0 | 38.8 | 71.9 | 96.9 | 97.6 | 11.0 | 37.5 | 52.1 |
| F&H + AUC [11] | 35.0 | 48.5 | 63.0 | 85.6 | 42.1 | 64.5 | 80.5 | 95.6 | 54.6 | 71.2 | 95.8 | 97.3 | 12.4 | 39.6 | 53.8 |
| Eban's [10] | 36.0 | 48.3 | 59.8 | 85.4 | 38.0 | 63.4 | 78.8 | 95.4 | 56.3 | 71.7 | 97.3 | 97.7 | 11.7 | 39.3 | 54.3 |
| Ours | 39.1 | 50.9 | 64.9 | 86.3 | 44.9 | 66.0 | 82.9 | 95.9 | 58.1 | 74.2 | 97.7 | 98.0 | 16.0 | 40.2 | 56.2 |

epochs. We also utilize the cosine schedule to adjust the learning rate. We also warm up the training for 5 epochs with the learning rate increasing from $10^{-4}$. The maximum number of epochs is 70, and the models that perform best on the validation set are saved for performance evaluation. The hyperparameters are set as follows: $\gamma = 40$ (Eq. (14)), $\sigma = 0.5$ (Eq. (15)), $N_c$ is set to 512 for the DR dataset and the AgePrediction dataset, and 1024 for UTKFace and NSFW.

**Evaluation metrics.** Aiming at maximizing the TPR@FPR metrics as shown in Eq. (3), we evaluate the TPR@FPR with $\alpha = 0.003, 0.01, 0.03$, respectively. Without prior knowledge, we consider all sub-problems as equally important, so $\lambda_k$ is set to 1.

**Competitors.** To validate the effectiveness of our proposed method, we compare with seven methods: Cross-Entropy (CE), NNRank [4], F&H [11], SoftLabel [9], F&H + AUC, DDAG [35] and Eban's method [10]. More descriptions on competitors are provided in Appendix D.

**Environments.** We implement our method and the competitors with modified *timm* [6] [40], and all the experiments are conducted on Ubuntu 16.04, Pytorch 1.7.0 [34], with a single NVIDIA RTX 3090 GPU.

## 4.3 Results and comparison

The quantitative results of the involved datasets are summarized in Tab. 2. It can be seen that our proposed method outperforms most of the involved methods on all datasets. Notice that on the dataset DR, with $\alpha = 0.003, 0.01, 0.03$, the proposed method achieves higher TPR than the best competitor's with 2.3%, 2.4% and 0.6%, respectively. Moreover, we have the following observations from the experimental results: **1)** Although the effectiveness of the competitors with regard to metrics like AUC, accuracy or mean square error has been verified in the literature, under more than half of the groups, these methods only achieve comparable performance to the baseline (CE). This phenomenon verifies that previous methods ignore the performance with a fixed low FPR, and this work is necessary; **2)** In the case of lower FPR, our method can bring more significant improvements, which is in line with the practical requirements: when more work is assigned to models instead of manpower, thresholds should be set to ensure that FPR is smaller to reduce the negative effects of model misjudgment. Therefore, TPR at a lower FPR is the performance when models are applied to a wider range, which is more instructive.

## 4.4 Sensitivity Analysis

There are two main hyperparameters in our proposed method, namely, $N_c$ to control the size of cache, and $\sigma$ dependent on the random walk assumption. In this sub-section, we analyze how these hyperparameters affect the model performance. We train models with a 2d grid search on $N_c \in \{64, 128, 256, 512, 1024, 2048\}$ and $\sigma \in \{0.1, 0.3, 0.5, 0.7, 1.0, 1.5, 2.0\}$. For each combination, we evaluate performance with $\alpha = 0.003, 0.01, 0.03$. The experiments are conducted on the DR dataset.

---

[6] https://github.com/rwightman/pytorch-image-models. Licensed Apache 2.0.

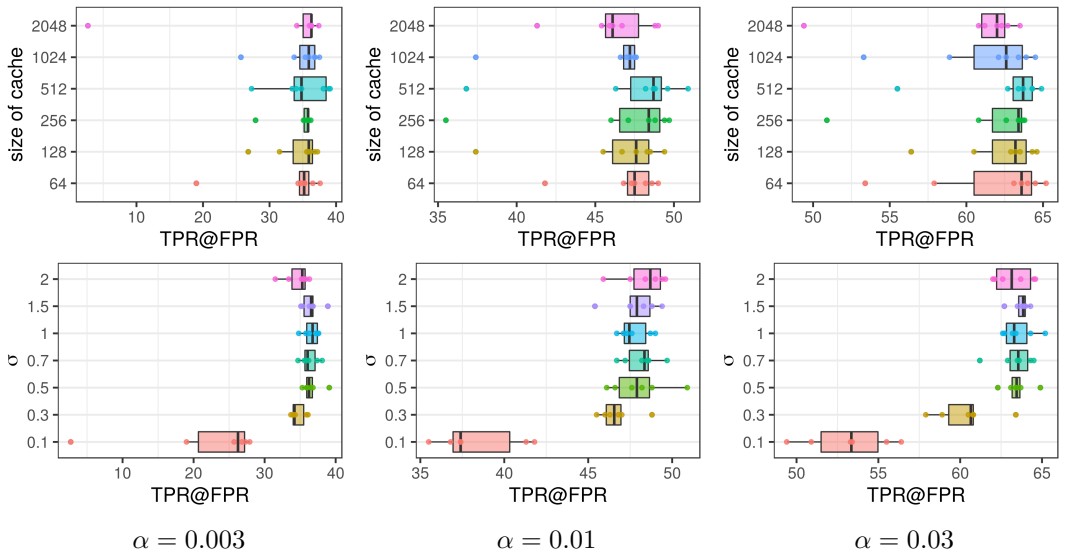

Figure 2: Sensitive analysis on $N_c$ and $\sigma$. Best viewed in color.

The results for $N_c$ are shown in the first row of Fig. 2. We can observe that too large or too small cache size will reduce model performance. This is due to a small cache uses less historical information, while a cache module that is too large will lead to a larger feature drift. We also observe that for $\alpha = 0.003$, $N_c$ has less impact on the results.

The second row of Fig. 2 exhibits the results for $\sigma$, from which we can see that extremely small $\sigma$ significantly reduces model performance. Directly optimizing the original problem can be viewed as $\sigma \rightarrow 0$, which verifies the conclusion we draw that directly optimizing the original is infeasible.

## 5   Limitations

Although the TPR@FPR metric shows practical significance and our proposed method performs well on this metric, they still have some limitations as follows.

**Require sufficient clear test data.** Since we focus on low FPRs, it requires sufficient clear data for testing. Otherwise, take $\alpha = 0.01$ as an example, if the number of negative examples is less than 100, only one false positive is allowed, which makes the metric sensitive to noises.

**Focus only on computer vision tasks.** The basic framework of our proposed method can be generalized to general multipartite ranking applications, but we focus on computer vision tasks in our experiments currently. This is because computer vision tasks are more diverse and challenging, and it is also one of the main areas of deep learning application, which is more suitable for our framework. We will consider extending it to other areas in future work.

## 6   Summary

In this paper, we study the TPR at a fixed FPR for multipartite ranking. From the practical perspective, we demonstrate the necessity of this metric in some scenarios sensitive to FPRs. Since TPR@FPR is a constrained optimization problem which is hard to solve in an end-to-end manner, we propose a novel method *CBA-CR* to approximately optimize this problem. On one hand, to tackle the challenge that the gradients of most negative examples are not available, we propose a relaxed form according to a random walk hypothesis. On the other hand, to decompose the objective into a sum of independent instance-wise terms, we approximate the rankings by caching features in the past iterations. Experiments on three real-world benchmarks are provided to show the effectiveness of our proposed method.

## Acknowledgments

This work was supported in part by the National Key R&D Program of China under Grant 2018AAA0102003, in part by National Natural Science Foundation of China: 61931008, 61620106009, 61836002 and 61976202, in part by the Fundamental Research Funds for the Central Universities, in part by Youth Innovation Promotion Association CAS, in part by the Strategic Priority Research Program of Chinese Academy of Sciences, Grant No. XDB28000000, and in part by Alibaba Group through Alibaba Research Fellowship Program.

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
