# A Details of Derivation

In this subsection, we will provide the proof that the difference between Eq. (5) to Eq. (7) is upper bounded (Thm. 1), the ranking can be estimated by a Gaussian process based on Asmp. 2 (Prop. 1), and rankings of instances in caches can be approximated with features in past iterations (Prop. 2).

## A.1 Proof of Theorem 1

The following lemmas are introduced to prove Thm. 1.

**Lemma 1.** *(Hölder's Inequality).* $\forall 0 < p < 1, q = 1 - p$, we have

$$\mathbb{E}[|XY|] \leq \mathbb{E}[|X|^{1/p}]^p \cdot \mathbb{E}[|Y|^{1/q}]^q.$$

**Lemma 2.** *Given a second-order derivable function $f$ and two positive arrays $\{a_i\}_i^n, \{b_i\}_i^n$, s.t. $a_i \geq b_i$, denote*

$$G_f(x) = \sum_i f(a_i \cdot x) / \sum_i f(b_i \cdot x),$$

$G_f(x)$ *is monotonically decreasing w.r.t. $x$ if $f$ is monotonic and $(f')^2 - f''f \geq 0$.*

*Proof.*

$$\left(\frac{f}{f'}\right)' = \frac{(f')^2 - f''f}{(f')^2} \geq 0$$

$$\Rightarrow \quad f(b_i \cdot x) \cdot f'(a_j \cdot x) \leq f(a_j \cdot x) \cdot f'(b_i \cdot x),$$

$$\Rightarrow \quad G'_f = \frac{1}{S^2} \left( \sum_i f'(a_i \cdot x) \sum_j f(b_j \cdot x) - \sum_i f(a_i \cdot x) \sum_j f'(b_j \cdot x) \right)$$

$$= \frac{1}{S^2} \sum_{i,j} \left( f'(a_i \cdot x) f(b_j \cdot x) - f(a_i \cdot x) f'(b_j \cdot x) \right) \leq 0,$$

$$\text{where} \quad S = \sum_j f(b_j \cdot x).$$

$\square$

*Proof of Thm. 1.*

On one hand, it can be proved that $\hat{\mathcal{R}}_p^\ell - \hat{\mathcal{R}}^\ell \leq 0$:

$$\hat{\mathcal{R}}_p^\ell - \hat{\mathcal{R}}^\ell = \frac{1}{|\mathcal{D}^+|} \sum_{\boldsymbol{x}_i^+ \in \mathcal{D}^+} \sum_{\boldsymbol{x}_i^- \in \mathcal{D}^-} \ell_{ij} p_j - \frac{1}{|\mathcal{D}^+|} \sum_{\boldsymbol{x}_i^+ \in \mathcal{D}^+} \ell_{i\beta}$$

$$= 1|\mathcal{D}^+| \sum_{\boldsymbol{x}_i^+, \boldsymbol{x}_i^- \in \mathcal{I}_\beta} \ell_{ij} p_j - \frac{1}{|\mathcal{D}^+|} \sum_{\boldsymbol{x}_i^+ \in \mathcal{D}^+} (1 - p_\beta) \ell_{i\beta}$$

$$= (|\mathcal{D}^-| - 1) \frac{1}{|\mathcal{I}_\beta|} \sum_{\boldsymbol{x}_i^+, \boldsymbol{x}_i^- \in \mathcal{I}_\beta} \ell_{ij} p_j - \frac{1}{|\mathcal{D}^+|} \sum_{\boldsymbol{x}_i^+ \in \mathcal{D}^+} (1 - p_\beta) \ell_{i\beta}$$

$$= (|\mathcal{D}^-| - 1) \hat{\mathbb{E}}_{\boldsymbol{x}^+, \boldsymbol{x}^- \in \mathcal{I}_\beta} [\ell \cdot p] - (1 - p_\beta) \hat{\mathbb{E}}_{\boldsymbol{x}^+ \in \mathcal{D}^+} [\ell_{i\beta}] \tag{18}$$

According to our sufficient condition, $\exists u \in (0, 1), v = 1 - u$, s.t.

$$(|\mathcal{D}^-| - 1)(\hat{\mathbb{E}}_{\boldsymbol{x}^+, \boldsymbol{x}^- \in \mathcal{I}_\beta} [\ell^{1/u}])^u (\hat{\mathbb{E}}_{\boldsymbol{x}^- \in \mathcal{D}_\beta^-} [p^{1/v}])^v \leq (1 - p_\beta) \hat{\mathbb{E}}_{\boldsymbol{x}^+ \in \mathcal{D}^+} [\ell_{i\beta}]. \tag{19}$$

With Lem. 1, we have

$$\hat{\mathcal{R}}_p^\ell - \hat{\mathcal{R}}^\ell \leq (|\mathcal{D}^-| - 1) \hat{\mathbb{E}}_{\boldsymbol{x}^+, \boldsymbol{x}^- \in \mathcal{I}_\beta} [\ell^{1/u}]^u \hat{\mathbb{E}}_{\boldsymbol{x}^- \in \mathcal{D}_\beta^-} [p^{1/v}]^v - (1 - p_\beta) \hat{\mathbb{E}}_{\boldsymbol{x}^+ \in \mathcal{D}^+} [\ell_{i\beta}] \leq 0. \tag{20}$$

On the other hand, $\hat{\mathcal{R}}_p^\ell/\hat{\mathcal{R}}^\ell$ is lower bounded by a monotonically decreasing function $g(\sigma)$, which tends to 1 when $\sigma \to 0$:

$$
\begin{aligned}
\hat{\mathcal{R}}_p^\ell/\hat{\mathcal{R}}^\ell =& \frac{\sum_{\boldsymbol{x}_i^+ \in \mathcal{D}^+, \boldsymbol{x}_j^- \in \mathcal{D}^-} \ell(s_i^+ - s_j^-)p_j}{\sum_{\boldsymbol{x}_i^- \in \mathcal{D}^-} \ell(s_i^+ - s_\beta^-)} \\
\geq& \frac{\sum_{\boldsymbol{x}_i^+ \in \mathcal{D}^+} \ell\left(\sum_{\boldsymbol{x}_j^- \in \mathcal{D}^-} (s_i^+ - s_j^-)p_j\right)}{\sum_{\boldsymbol{x}_i^- \in \mathcal{D}^-} \ell(s_i^+ - s_\beta^-)} \\
=& \frac{\sum_{\boldsymbol{x}_i^+ \in \mathcal{D}^+} \ell\left(\bar{\delta}_i \cdot \sigma\right)}{\sum_{\boldsymbol{x}_i^- \in \mathcal{D}^-} \ell(\delta_{ij} \cdot \sigma)} \\
\triangleq& g(\sigma).
\end{aligned}
\tag{21}
$$

We have $\bar{\delta}_i - \delta_{ij} = (s_i^- - \sum_{\boldsymbol{x}_j \in \mathcal{D}^-} s_j^- p_j)\sigma \geq 0$ due to condition *(b)*. According to $\ell(0) > 0$ and Lem. 2, $g(\sigma)$ is monotonically decreasing, and $g(\sigma) \to 1$ when $\sigma \to 0$. Since $\hat{\mathcal{R}}_p^\ell/\hat{\mathcal{R}}^\ell \leq 1$, we have $\hat{\mathcal{R}}_p^\ell/\hat{\mathcal{R}}^\ell \to 1$.

$\square$

Thm. 1 shows that $\hat{\mathcal{R}}_p^\ell$ could be an effective estimation of $\hat{\mathcal{R}}^\ell$, especially when the score function $f$ converges. Thus, minimizing Eq. (7) could also minimize Eq. (5).

Next, we discuss the three hypotheses. We choose logistic function as our surrogate function, i.e., $\ell(x) = \log(1 + e^{-x})$, which meets the condition *(a)*: $(\ell^2) - \ell''\ell = \left(1 - e^x \cdot \log(1 + e^{-x})\right)/(1 + e^x)^2 \geq 0$. What's more, an intuitive explanation of condition *(b)* is that the weighted average of all negative scores is less than the $\beta$-largest score, which holds when $\beta$ is small.

To verify that the condition *(c)* is easy to satisfy, we also provide an experiment on synthetic data. Specifically, the scores are sampled from Gaussian distribution: $\{s_i^+\}_i \overset{i.i.d}{\sim} \mathcal{N}(0.7, \sigma)$, $\{s_i^-\}_i \overset{i.i.d}{\sim} \mathcal{N}(0.3, \sigma)$, and we show how $\hat{\mathcal{R}}^\ell$ and $\hat{\mathcal{R}}_p^\ell$ change with $\sigma$ decreasing in Fig. 3, from which could be told that $\hat{\mathcal{R}}^\ell$ and $\hat{\mathcal{R}}_p^\ell$ tends to be consistent with $f$ converging.

### A.2 Proof of Proposition 1

**Restate of Proposition 1.** *Assume $\Delta t \to 0$, $\Delta s \to 0$, and $\Delta s^2 = \beta \Delta t$, the probability that the ranking of $\boldsymbol{x}$ reaches the ranking $\lceil \alpha|\mathcal{A}| \rceil$ after a period $T$*

$$
\mathbb{P}(\boldsymbol{x}; \mathcal{A}) \approx \frac{1}{\sqrt{2\pi}\sigma} \exp\left(-\frac{(R(\boldsymbol{x}; \mathcal{A}, \boldsymbol{\theta})/|\mathcal{A}| - \alpha)^2}{2\sigma^2}\right)
\tag{22}
$$

*where $\sigma = \sqrt{p\beta T/2}$.*

*Proof.* Consider a one-dimensional random walk process. A particle starts from 0, and walks $\Delta s$ distance to the left or right with probability $p/2$ per $\Delta t$ time, and stays in place with probability $1 - p$. The probability of reaching $k\Delta s$ after $n = T/\Delta t$ steps is denoted as $p_k$. Define a generating function $G_n(z)$ of $p_k$ as follows:

$$
G_n(z) = \sum_{k=-n}^{n} p_k z^{k\Delta s}
\tag{23}
$$

Obviously $G_0(z) = 1$. Consider the process from step $n - 1$ to step $n$, the particle walks left or right with probability $p/2$, which corresponds to $\frac{p}{2}z^{-\Delta s}G_{n-1}(z)$ and $\frac{p}{2}z^{\Delta s}G_{n-1}(z)$, respectively. Additionally, it stays the same with probability $1 - p$, corresponding to $(1 - p)G_{n-1}(z)$. Therefore, we have

$$
G_n(z) = (\frac{p}{2}(z^{\Delta s} + z^{-\Delta s}) + 1 - p)G_{n-1}(z) = (\frac{p}{2}(z^{\Delta s} + z^{-\Delta s}) + 1 - p)^n
\tag{24}
$$

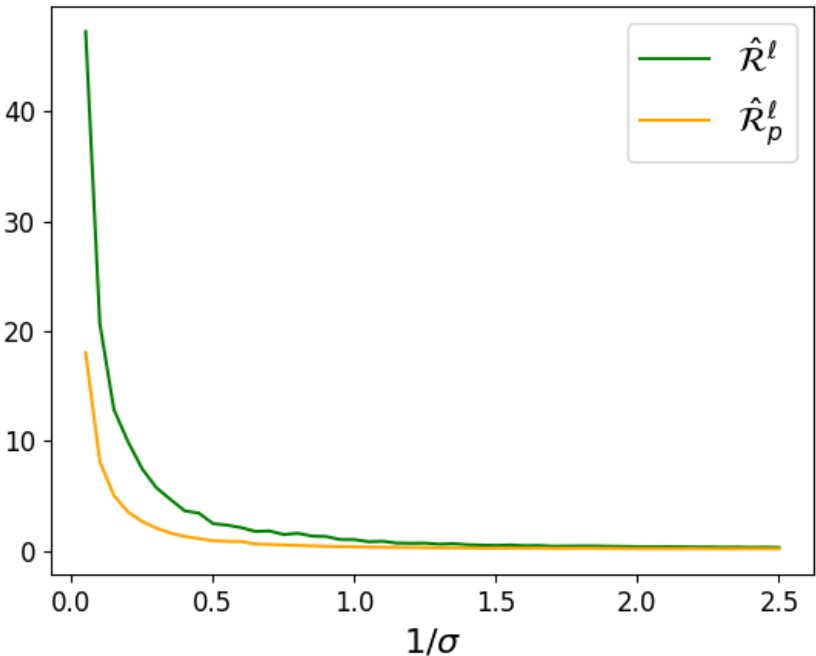

Figure 3: Empirical risk with/without the approximation. Best viewed in color.

when $\Delta s \to 0$, $\Delta t \to 0$, $\Delta s^2 = \beta \Delta t$, $z = e^{-i\omega}$, $G(z)$ is the Fourier transform of $p_k$:

$$
\begin{aligned}
G(z) &= \mathcal{F}\{p_k\} \\
&= (\frac{p}{2}(z^{\Delta s} + z^{-\Delta s}) + 1 - p)^n \\
&= (\frac{p}{2}(e^{i\omega\Delta s} + e^{-i\omega\Delta s}) + 1 - p)^n \\
&= (p\cos(\omega\Delta s) + 1 - p)^n \\
&= (p(1 - 2\sin^2(\frac{1}{2}\omega\Delta s)) + 1 - p)^n \\
&= (1 - 2p\sin^2(\frac{1}{2}\omega\Delta s))^n \\
&\overset{(1)}{\approx} (p(1 - \frac{1}{2}\omega^2\Delta s^2) + 1 - p)^n \\
&= (1 - \frac{p}{2}\omega^2\beta\Delta t)^{T/\Delta t} \\
&= (1 - \frac{p}{2}\omega^2\beta\Delta t)^{-1/(p\omega^2\beta\Delta t/2)\times(-pT\beta\omega^2/2)} \\
&\overset{(2)}{\approx} \exp(-\frac{p}{2}T\beta\omega^2)
\end{aligned}
\tag{25}
$$

Therefore, $p_k$ can be obtained by the inverse Fourier transform of $G$:

$$
\begin{aligned}
p_k &\approx \mathcal{F}^{-1}\{\exp(-\frac{p}{2}T\beta\omega^2)\} \\
&= \frac{1}{\sqrt{2\pi}\sigma}\exp(-\frac{k^2\Delta s^2}{2\sigma^2}) \\
&= \frac{1}{\sqrt{2\pi}\sigma}\exp(-\frac{x^2}{2\sigma^2})
\end{aligned}
\tag{26}
$$

where $\sigma = \sqrt{p\beta T/2}$, $x$ is the walk distance.

The original problem is equivalent to the probability that the walking distance is $R(\boldsymbol{x}; \mathcal{A}, \boldsymbol{\theta})/|\mathcal{A}| - \alpha$, thus we have

$$\mathbb{P}(\boldsymbol{x}; \mathcal{A}) \approx \frac{1}{\sqrt{2\pi}\sigma} \exp(-\frac{(R(\boldsymbol{x}; \mathcal{A}, \boldsymbol{\theta})/|\mathcal{A}| - \alpha)^2}{2\sigma^2}) \tag{27}$$

(1). It's due to $\sin(x) \approx x$ when $x \to 0$ and $\Delta s \to 0$, $(1+x)^n$ is a continuous smooth function.
(2). It's due to $(1+x)^{1/x} \approx e$ when $x \to 0$ and $\Delta t \to 0$.

$\square$

### A.3  Proof of Proposition 2

**Restate of Proposition 2.** *Assume that $g_\phi$ is L-Lipschitz continuous on $\phi$, and $\|\phi - \hat{\phi}\| \le \epsilon$, the error of ranking estimation on $\mathcal{A}$ is*

$$|R(\boldsymbol{x}; \mathcal{A}, \boldsymbol{\theta}, \boldsymbol{w}) - R(\boldsymbol{x}; \mathcal{A}, \hat{\boldsymbol{\theta}}, \boldsymbol{w})| \le \sum_{i \in \mathcal{A}} I[2\epsilon L > |\boldsymbol{w}^T(g_\phi(\boldsymbol{x}) - g_\phi(\boldsymbol{x}_i))|/\|\boldsymbol{w}\|] \tag{28}$$

*Proof.*

$$\begin{aligned} |\Delta R| &= |R(\boldsymbol{x}; \mathcal{A}, \boldsymbol{\theta}, \boldsymbol{w}) - R(\boldsymbol{x}; \mathcal{A}, \hat{\boldsymbol{\theta}}, \boldsymbol{w})| \\ &= |\sum_{i \in \mathcal{A}} I[f_{\boldsymbol{\theta}}(\boldsymbol{x}) < f_{\boldsymbol{\theta}}(\boldsymbol{x}_i)] - \sum_{i \in \mathcal{A}} I[f_{\hat{\boldsymbol{\theta}}}(\boldsymbol{x}) < f_{\hat{\boldsymbol{\theta}}}(\boldsymbol{x}_i)]| \\ &\le \sum_{i \in \mathcal{A}} |I[f_{\boldsymbol{\theta}}(\boldsymbol{x}) < f_{\boldsymbol{\theta}}(\boldsymbol{x}_i)] - I[f_{\hat{\boldsymbol{\theta}}}(\boldsymbol{x}) < f_{\hat{\boldsymbol{\theta}}}(\boldsymbol{x}_i)]| \\ &= \sum_{i \in \mathcal{A}} I[|\Delta f_{\boldsymbol{\theta}}(\boldsymbol{x}) - \Delta f_{\boldsymbol{\theta}}(\boldsymbol{x}_i)| > |f_{\boldsymbol{\theta}}(\boldsymbol{x}) - f_{\boldsymbol{\theta}}(\boldsymbol{x}_i)|] \end{aligned} \tag{29}$$

where $\Delta f_{\boldsymbol{\theta}}(\boldsymbol{x}) = f_{\boldsymbol{\theta}}(\boldsymbol{x}) - f_{\hat{\boldsymbol{\theta}}}(\boldsymbol{x})$. Since $g_\phi$ is L-Lipschitz continuous on $\phi$, *i.e.*, $\|g_\phi(\boldsymbol{x}) - g_{\hat{\phi}}(\boldsymbol{x})\| \le L\|\phi - \hat{\phi}\| \le \epsilon L$, we have

$$\begin{aligned} |\Delta f_{\boldsymbol{\theta}}(\boldsymbol{x})| &= |f_{\boldsymbol{\theta}}(\boldsymbol{x}) - f_{\hat{\boldsymbol{\theta}}}(\boldsymbol{x})| \\ &= |\boldsymbol{w}^T(g_\phi(\boldsymbol{x}) - g_{\hat{\boldsymbol{\theta}}}(\boldsymbol{x}))| \\ &\le \|w\|\|g_\phi(\boldsymbol{x}) - g_{\hat{\phi}}(\boldsymbol{x})\| \\ &\le \epsilon L\|w\| \end{aligned} \tag{30}$$

thus

$$|\Delta f_{\boldsymbol{\theta}}(\boldsymbol{x}) - \Delta f_{\boldsymbol{\theta}}(\boldsymbol{x}_i)| \le 2\epsilon L\|w\| \tag{31}$$

From Eq. (29) and Eq. (30), we have

$$\begin{aligned} |\Delta R| &\le \sum_{i \in \mathcal{A}} I[2\epsilon L\|w\| > |f_{\boldsymbol{\theta}}(\boldsymbol{x}) - f_{\boldsymbol{\theta}}(\boldsymbol{x}_i)|] \\ &= \sum_{i \in \mathcal{A}} I[2\epsilon L > |\boldsymbol{w}^T(g_\phi(\boldsymbol{x}) - g_\phi(\boldsymbol{x}_i))|/\|\boldsymbol{w}\|] \end{aligned} \tag{32}$$

$\square$

## B  Hypothesis Testing

In this subsection, a hypothesis test for Assumption 1 is conducted. Specifically, during the training phase, we randomly select 1000 instances $\{\boldsymbol{x}_i\}_{i=1}^{1000}$ from the DR dataset, and track the difference sequence $\{d^{(t)} = \mathbb{E}[R(\boldsymbol{x}; \mathcal{A}, \boldsymbol{\theta}_{t+1}) - R(\boldsymbol{x}; \mathcal{A}, \boldsymbol{\theta}_t)]\}_t$. The null hypothesis $\mathcal{H}_0$ and the alternative hypothesis $\mathcal{H}_1$ are as follows:

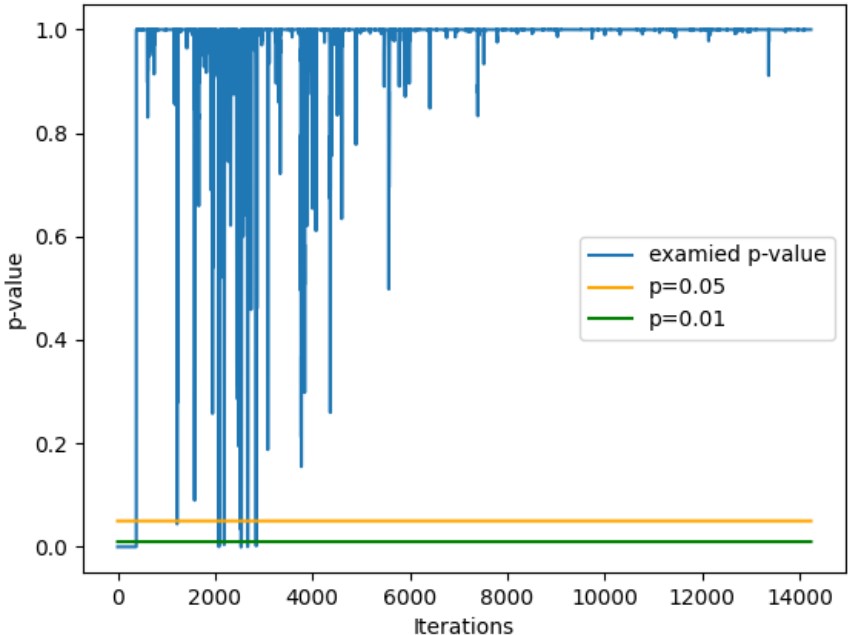

Figure 4: Examined p-value on $\hat{\chi}^2(m)$. Best viewed in color.

$\mathcal{H}_0$ : the difference sequence $\{d^{(t)}\}$ is white noise at a short period.
$\mathcal{H}_1$ : the difference sequence $\{d^{(t)}\}$ is not white noise at a short period.

We apply the Box-Pierce test which is commonly used in signal processing. Formally, the examined variable is constructed as

$$\hat{\chi}^2(m) \stackrel{\triangle}{=} N(\hat{\rho}_1^2 + \hat{\rho}_2^2 + \cdots + \hat{\rho}_m^2) \qquad (33)$$

where $m$ is the stage to examined, $N$ is the number of samples and $\hat{\rho}_k$ is the estimator of the $k$-rank variance of $\{d^{(t)}\}$. According to $\mathcal{H}_0$, $\hat{\chi}^2(m)$ approximately obeys a $\chi^2(m)$ distribution. Therefore, we can transform the obtained $\hat{\chi}^2(m)$ into the p-value, which is visualized in Fig. 4. From the visualization results, it can be observed that as the training stabilizes, the p-value is much greater than 0.05 in most cases, so there is not enough evidence to reject $\mathcal{H}_0$.

## C   Algorithm Summary

In this subsection, we provide a summary for the process of our proposed algorithm in Alg. 1. Firstly, we initialize the model parameters and the cache (`Line 1 and 2`). At each iteration, we extract embeddings for the sampled mini-batch (`Line 5`), and update the cross-batch cache with extracted embeddings (`Line 6`). Then, we calculate the rankings in cache (`Line 7`). Afterward, the rankings of instances in mini-batch are estimated, and transformed into $\tilde{\mathbb{P}}(\boldsymbol{x}_i^{k-}; \mathcal{D}_k^-)$ (`Line 8 and 9`). Finally, the model parameters $\boldsymbol{\theta}$ can be updated with gradient backpropagation (`Line 10`).

## D   Details of Competitors

To validate the effectiveness of our proposed method, we compare with seven methods: Cross-Entropy (CE), NNRank [4], F&H [11], F&H + AUC, SoftLabel [9], DDAG [35] and Eban's method [10]. For fair comparison, we replace the feature extractor with ReXNet200 for all competitors, and train these