# OpenReview forum: "When False Positive is Intolerant: End-to-End  Optimization with Low FPR for Multipartite Ranking"
_NeurIPS.cc/2021/Conference — NeurIPS 2021 Poster_

### Official Review · Reviewer_EgXj · 2021-07-10

**Rating:** 5
**Confidence:** 3

**Summary:**

In this paper, the authors studied a particular metric for M-class multipartite ranking, with the particular motivation of FPR-sensitive applications in mind. Given that the combinatorial objective is hard to solve exactly, a series of relaxations are introduced / assumptions brought in for the algorithm to be amenable to mini-batch gradient updates. The high-level approach taken is to turn the problem into M-1 bipartite ranking sub-problems, after which a probability-based differentiable surrogate loss function are used and a cache module are leveraged to efficiently approximate the ranking of examples. Numerical experiments are also conducted on a series of real-world datasets.

**Limitations And Societal Impact:**

Yes.

**Main Review:**

 While the assumptions made in the paper sound reasonable to me, it would be good (and important) to have an approximation guarantee on the metric of interest in my opinion - it is unclear to me what can be said about Definition 2 (eqn 3) after the whole sequence of relaxations. It would also be interesting to trace out how much each level of approximation improve the performance / runtime on real data (going from (5) to (6) for example, and similarly for the ones in section 3.4). The relevant literature on ranking is adequately surveyed and the claims do look right.

The experimental results are shown, but it seems that some of the margin against other baselines are not that significant and makes me wonder what is the runtime comparison for those methods. On the other hand, from a more practical perspective, do the authors find other ways to estimate the probability weighting improving the performance in practice, beyond the random walk hypothesis made? These seem like questions that would be nice to investigate further.

Minor typos: line 73: firstly -> first

**Time Spent Reviewing:**

3

---

> ### Author Response · Authors · 2021-08-10
> **Response to Reviewer EgXj**
>
> Thanks for your time and constructive comments on our paper. We would like to discuss the following issues:
>
> ### **Q1**:
> Approximation guarantee.
> ### **A1**:
> The approximation guarantee on ranking estimation is provided in Prop. 2. To discuss the approximation guarantee from Eq. (5) to Eq. (6), we provide a sufficient condition under which the objectives are consistent with the performance increasing. Here we consider a sub-problem and ignore the subscript $k$. Denote $\hat{\mathcal{R}^\ell_p}$ as the empirical risk of TPR@FPR (as defined in Eq. (5)), and $\hat{\mathcal{R}^\ell_p}$ as the empirical risk of the approximated version (as defined in Eq. (6)), i.e.,
>
> &emsp;&emsp; $\hat{\mathcal{R}^\ell} = \frac{1}{|\mathcal{D}^+|}\sum\limits_{\mathbf{x}_i^+\in |\mathcal{D}^-|} \ell(s_i^+ - s_j^-)$ ,
>
> &emsp;&emsp; $\hat{\mathcal{R}^\ell_p} = \frac{1}{|\mathcal{D}^+|}\sum\limits_{\mathbf{x}_i^+\in |\mathcal{D}^+|, \textbf{x}_j^-\in |\mathcal{D}^-|} \ell(s_i^+ - s_j^-)\mathbb{P}(\mathbf{x}_j^-; |\mathcal{D}^-|)$,
>
> we provide a sufficient condition for $\hat{\mathcal{R}^\ell_p} \rightarrow \hat{\mathcal{R}^\ell}$ in the following theorem. **The proof is shown in the next comment.**
>
> **Theorem 1.** *Given a probability estimation $\mathbb{P}(\cdot; |\mathcal{D}^-|): \mathbb{R} \mapsto (0, 1)$, and a surrogate loss function $\ell(\cdot)$, denote $\ell_{ij} = \ell(s_i^+ - s_j^-), p_j = \mathbb{P}(\mathbf{x}_j^-;\mathcal{D}^-), \beta = \lceil \alpha |\mathcal{D}^-| \rceil$, $\mathcal{D}^-_\beta=\mathcal{D}^-\backslash$ {$ \mathbf{x}_\{(\beta)\}$}, and $\mathcal{I}_\beta = \mathcal{D}^+ \times \mathcal{D}^-_\beta$; denote $\hat{\mathbb{E}}_\{\mathbf{x}^+ \in \mathcal{D}^+\} [z]$ as the empirical expectation of $z$ over $\mathcal{D}^+$, and similarly for $\hat{\mathbb{E}}_\{\mathbf{x}^- \in \mathcal{D}^-_\beta\} [z]$, $\hat{\mathbb{E}}_\{\mathbf{x}^+, \mathbf{x}^- \in \mathcal{I}_\beta\} [z]$; denote $\sigma = Var_\{\mathbf{x}_i^+\in\mathcal{D}^+, \mathbf{x}_j^-\in\mathcal{D}^-\}[s_i^+ - s_j^-]$, $\delta_\{ij\} = (s_i^+ - s_j^-) / \sigma$, $\bar{\delta_i} = (s_i^+ - \sum\limits_\{\mathbf{x}_j^- \in \mathcal{D}^-\}s_j^-p_j)/\sigma$.
> Assume $\sum\limits_\{\mathbf{x}_j^- \in \mathcal{D}^-\} p_j = 1$. We have $\hat{\mathcal{R}^\ell_p} \rightarrow \hat{\mathcal{R}^\ell}$ when $\sigma \rightarrow 0$ if the following conditions are met:*
>
> &emsp;&emsp; *(a) $\ell$ is convex, monotonically decreasing, $\ell(0) > 0$, and $(\ell')^2 - \ell''\cdot \ell \geq 0$;*
>
> &emsp;&emsp; *(b) $\sum\limits_{\mathbf{x}_j^- \in \mathcal{D}^-}s_j^-p_j \leq s_\beta^- $;*
>
> &emsp;&emsp; *(c) $\inf_\{u\in(0,1),v=1-u\}[A_u - B_v] \leq 0$, where*
>
> &emsp;&emsp;&emsp; *$A_u = ({|\mathcal{D}^-| - 1})(\hat{\mathbb{E}}_{\mathbf{x}^+,\mathbf{x}^- \in \mathcal{I}_\beta}[\ell^{1/u}])^{u},$*
>
> &emsp;&emsp;&emsp; *$B_v = (1 - p_\beta) \hat{\mathbb{E}}_\{\mathbf{x}^+ \in \mathcal{D}^+\}[\ell_\{i\beta\}] / (\hat{\mathbb{E}}_\{\mathbf{x}^- \in \mathcal{D}^-_\beta\}[p^{1/v}])^{v}.$*
>
> Thm. 1 shows that $\hat{\mathcal{R}^\ell_p}$ could be an effective estimation of $\hat{\mathcal{R}^\ell}$, especially when the score function $f$ converges. Thus, minimizing Eq. (6) could also minimize Eq. (5) when $\sigma$ goes to zero, i.e., both the positive examples and negative examples have zero variance.
>
> Next, we discuss the three hypotheses. We choose logistic function as our surrogate function, i.e., $\ell(x) = \log(1 + e^{-x})$, which meets the condition *(a)*: $(\ell^2) - \ell''\ell = \left(1 - e^x\cdot\log(1 + e^{-x})\right) / (1 + e^x)^2 \geq 0$. What's more, an intuitive explanation of condition *(b)* is that the weighted average of all negative scores is less than the $\beta$-largest score, which holds considering $\beta$ is often small.
>
> To verify  condition *(c)*, we also provide an experiment on synthetic data. Specifically, the scores are sampled from Gaussian distribution: {$s_i^+$ } $_i \stackrel{i.i.d}{\sim}\mathcal{N}(0.7, \sigma)$, \{$s_i^-$\}$_i \stackrel{i.i.d}{\sim}\mathcal{N}(0.3, \sigma)$, and we show in Fig. 4 how $\hat{\mathcal{R}^\ell}$ and $\hat{\mathcal{R}^\ell_p}$ change with $\sigma$ decreasing. It could be told that $\hat{\mathcal{R}^\ell}$ and $\hat{\mathcal{R}^\ell_p}$ tend to be consistent with $\sigma$ goes to zero.
>
> Link to Fig. 4: <https://ibb.co/pQSPJZc>
>
> &nbsp;
>
> ### **Q2**:
> How much does each level of approximation improve the performance/runtime from (5) to (6), and cross-batch cache?
> ### **A2**:
> One of the main reasons for proposing Eq. (6) is that optimizing Eq. (5) in a mini-batch manner is infeasible. We tested the time consumption of selecting negative examples in a full-batch manner: about 173 min per epoch, while the approximated version only needs 5 min per epoch. Moreover, even if we introduce the caching mechanism to speed up negative sample selection, the training processing still fails to converge if we use Eq. (5) directly since it ignores most of the negative examples.
>
> &nbsp;
>
> ### **Q3**:
> The runtime comparison to baselines.
> ### **A3**:
> The inference time of all methods is the same due to the same network architecture. Moreover, under the default setting, we list the training time of those methods as follows:
>
> --------------------------------training-time-for-an-iteration-(ms)---------------------------------
>
> |CE|NNRank|F&H|DDAG|SoftLabel|F&H+AUC|Eban’s|ours|
> |:---:|:---:|:---:|:---:|:---:|:---:|:---:|:---:|
> |74|74|74|81|74|74|78|84|
>
> The proposed method requires a slightly longer training time, but the gap is not obvious.
>
> &nbsp;
>
> ### **Q4**:
> Other ways to estimate the probability weighting improving the performance in practice, beyond the random walk hypothesis made.
> ### **A4**:
> As shown in Thm. 1, the probability weighting strategy is preferred as long as the three conditions are met. Despite the implementation of weighting could be varied, the proposed weighting based on the random walk hypothesis is observed from experiments and is supported by hypothesis test (please refer to Appendix B).

---

> ### Author Response · Authors · 2021-08-10
> **Proof of Theorem 1**
>
> ### **Proof of Theorem 1**
>
> The following lemmas are introduced to prove Thm. 1.
>
> **Lemma 1.** (H\"older’s Inequality)
> $\forall 0 < p < 1, q = 1 - p$, we have
>
> &emsp;&emsp;&emsp; $\mathbb{E}[|XY|] \leq \mathbb{E}[|X|^{1/p}]^p \cdot \mathbb{E}[|Y|^{1/q}]^q.$
>
> **Lemma 2.**
> Given a second-order derivable function $f$ and two positive arrays {$a_i$}$_i^n$, {$b_i$}$_i^n$, s.t. $a_i \geq b_i$, denote
>
> &emsp;&emsp;&emsp; $G_f(x) = \sum_i f(a_i \cdot x) / \sum_i f(b_i \cdot x),$
>
> $G_f(x)$ is monotonically decreasing w.r.t. $x$ if $f$ is monotonic and $(f')^2 - f''f\geq 0$.
>
> *Proof.*
>
> &emsp;&emsp;&emsp;&emsp; $\left(\frac{f}{f'}\right)' = \frac{(f')^2 - f''f}{(f')^2} \geq 0$
>
> &emsp;&emsp;&emsp; $\Rightarrow f(b_i \cdot x)\cdot f'(a_j \cdot x) \leq f(a_j \cdot x)\cdot f'(b_i \cdot x),$
>
> &emsp;&emsp;&emsp; $\Rightarrow G_f' = \frac{1}{S^2}\left(\sum_{i} f'(a_i \cdot x) \sum_{j}f(b_j \cdot x) - \sum_{i} f(a_i \cdot x) \sum_{j}f'(b_j \cdot x)\right)$
>
> &emsp;&emsp;&emsp;&emsp;&emsp;&emsp; $ = \frac{1}{S^2}\sum_{i,j} \left( f'(a_i \cdot x)f(b_j \cdot x) - f(a_i \cdot x)f'(b_j \cdot x)\right) \leq 0,$
>
> where &emsp; $S = \sum_j f(b_j \cdot x).$
>
> &emsp;&emsp;&emsp;&emsp;&emsp;&emsp;&emsp;&emsp;&emsp;&emsp;&emsp;&emsp;&emsp;&emsp;&emsp;&emsp;&emsp;&emsp;&emsp;&emsp;&emsp;&emsp; Q.E.D
>
> *Proof of Theorem 1.*
>
> On one hand, it can be proved that $\hat{\mathcal{R}^\ell_p} - \hat{\mathcal{R}^\ell} \leq 0$:
>
> &emsp;&emsp;&emsp;&emsp; $\hat{\mathcal{R}^\ell_p} - \hat{\mathcal{R}^\ell}$
>
> &emsp;&emsp;&emsp; $=\frac{1}{|\mathcal{D}^+|}\sum_\{\mathbf{x}_i^+\in \mathcal{D}^+\}\sum_\{\mathbf{x}_i^-\in \mathcal{D}^-\} \ell_\{ij\}p_j - \frac{1}{|\mathcal{D}^+|}\sum_\{\mathbf{x}_i^+\in \mathcal{D}^+\}\ell_\{i\beta\}$
>
> &emsp;&emsp;&emsp; $={|\mathcal{D}^+|}\sum_\{\mathbf{x}_i^+,\mathbf{x}_i^- \in \mathcal{I}_\beta\}\ell_\{ij\}p_j - \frac{1}{|\mathcal{D}^+|}\sum_\{\mathbf{x}_i^+\in \mathcal{D}^+\}(1 - p_\beta)\ell_\{i\beta\}$
>
> &emsp;&emsp;&emsp; $={(|\mathcal{D}^-| - 1)}\frac{1}{|\mathcal{I}_\beta|}\sum_\{\mathbf{x}_i^+,\mathbf{x}_i^- \in \mathcal{I}_\beta\}\ell_\{ij\}p_j - \frac{1}{|\mathcal{D}^+|}\sum_\{\mathbf{x}_i^+\in \mathcal{D}^+\}(1 - p_\beta)\ell_\{i\beta\}$
>
> &emsp;&emsp;&emsp; $={(|\mathcal{D}^-| - 1)}\hat{\mathbb{E}}_\{\mathbf{x}^+, \mathbf{x}^- \in \mathcal{I}_\beta\}[\ell\cdot p] - (1 - p_\beta) \hat{\mathbb{E}}_\{\mathbf{x}^+ \in \mathcal{D+}\} [\ell_\{i\beta\}]$
>
> According to our sufficient condition *(c)*, $\exists u\in (0,1), v = 1-u$, s.t.
>
> &emsp;&emsp;&emsp; ${(|\mathcal{D}^-| - 1)}(\hat{\mathbb{E}}_\{\mathbf{x}^+,\mathbf{x}^- \in \mathcal{I}_\beta\}[\ell^{1/u}])^{u}(\hat{\mathbb{E}}_\{\mathbf{x}^- \in \mathcal{D}^-_\beta\}[p^{1/v}])^{v} \leq (1 - p_\beta) \hat{\mathbb{E}}_\{\mathbf{x}^+ \in \mathcal{D}^+\}[\ell_\{i\beta\}]. $
>
> With Lem. 1, we have
>
> &emsp;&emsp;&emsp; $\hat{\mathcal{R}^\ell_p} - \hat{\mathcal{R}^\ell} \leq {(|\mathcal{D}^-| - 1)}\hat{\mathbb{E}}_\{\mathbf{x}^+, \mathbf{x}^- \in \mathcal{I}_\beta\}[\ell^{1/u}]^{u}\hat{\mathbb{E}}_\{\mathbf{x}^- \in \mathcal{D}^-_\beta\}[p^{1/v}]^{v} - (1 - p_\beta) \hat{\mathbb{E}}_\{\mathbf{x}^+ \in \mathcal{D+}\} [\ell_\{i\beta\}] \leq 0.$
>
>
> On the other hand, $\hat{\mathcal{R}^\ell_p} / \hat{\mathcal{R}^\ell}$ is lower bounded by a monotonically decreasing function $g(\sigma)$, which approaches to $1$ when $\sigma \rightarrow 0$:
>
> &emsp;&emsp;&emsp; $\hat{\mathcal{R}^\ell_p} / \hat{\mathcal{R}^\ell} $
>
> &emsp;&emsp;&emsp; $= \frac{\sum_\{\mathbf{x}_i^+\in \mathcal{D}^+, \mathbf{x}_j^-\in \mathcal{D}^-\}\ell(s_i^+ - s_j^-)p_j}{\sum_\{\mathbf{x}_i^-\in \mathcal{D}^-\}\ell(s_i^+ - s_\beta^-)}$
>
> &emsp;&emsp;&emsp; $\geq \frac{\sum_\{\mathbf{x}_i^+\in \mathcal{D}^+\}\ell\left(\sum_\{\mathbf{x}_j^-\in \mathcal{D}^-\}(s_i^+ - s_j^-)p_j\right)}{\sum_\{\mathbf{x}_i^-\in \mathcal{D}^-\}\ell(s_i^+ - s_\beta^-)}$
>
> &emsp;&emsp;&emsp; $= \frac{\sum_\{\mathbf{x}_i^+\in \mathcal{D}^+\}\ell\left(\bar{\delta_i} \cdot \sigma\right)}{\sum_\{\mathbf{x}_i^-\in \mathcal{D}^-\}\ell(\delta_\{ij\} \cdot \sigma)}$
>
> &emsp;&emsp;&emsp; $\triangleq g(\sigma).$
>
> The inequality is due to Jensen's inequality. Moreover, we have $\bar{\delta_i} - \delta_\{ij\} = (s_i^- - \sum_\{\mathbf{x}_j \in \mathcal{D}^-\}s_j^-p_j) \sigma \geq 0$ due to condition *(b)*. According to $\ell(0) > 0$ and Lem. 2, $g(\sigma)$ is monotonically decreasing, and $g(\sigma) \rightarrow 1$ when $\sigma \rightarrow 0$. Since $\hat{\mathcal{R}^\ell_p} / \hat{\mathcal{R}^\ell} \leq 1,$ we have $\hat{\mathcal{R}^\ell_p} / \hat{\mathcal{R}^\ell} \rightarrow 1$.
>
> &emsp;&emsp;&emsp;&emsp;&emsp;&emsp;&emsp;&emsp;&emsp;&emsp;&emsp;&emsp;&emsp;&emsp;&emsp;&emsp;&emsp;&emsp;&emsp;&emsp;&emsp;&emsp; Q.E.D

---

> ### Author Response · Authors · 2021-08-10
> **Response to Reviewer EgXj**
>
> Thanks for your time and constructive comments on our paper. We would like to discuss the following issues:
>
> ### **Q1**:
> Approximation guarantee.
> ### **A1**:
> The approximation guarantee on ranking estimation is provided in Prop. 2. To discuss the approximation guarantee from Eq. (5) to Eq. (6), we provide a sufficient condition under which the objectives are consistent with the performance increasing. Here we consider a sub-problem and ignore the subscript $k$. Denote $\hat{\mathcal{R}^\ell_p}$ as the empirical risk of TPR@FPR (as defined in Eq. (5)), and $\hat{\mathcal{R}^\ell_p}$ as the empirical risk of the approximated version (as defined in Eq. (6)), i.e.,
>
> &emsp;&emsp; $\hat{\mathcal{R}^\ell} = \frac{1}{|\mathcal{D}^+|}\sum\limits_{\mathbf{x}_i^+\in |\mathcal{D}^-|} \ell(s_i^+ - s_j^-)$ ,
>
> &emsp;&emsp; $\hat{\mathcal{R}^\ell_p} = \frac{1}{|\mathcal{D}^+|}\sum\limits_{\mathbf{x}_i^+\in |\mathcal{D}^+|, \textbf{x}_j^-\in |\mathcal{D}^-|} \ell(s_i^+ - s_j^-)\mathbb{P}(\mathbf{x}_j^-; |\mathcal{D}^-|)$,
>
> we provide a sufficient condition for $\hat{\mathcal{R}^\ell_p} \rightarrow \hat{\mathcal{R}^\ell}$ in the following theorem. **The proof is shown in the next comment.**
>
> **Theorem 1.** *Given a probability estimation $\mathbb{P}(\cdot; |\mathcal{D}^-|): \mathbb{R} \mapsto (0, 1)$, and a surrogate loss function $\ell(\cdot)$, denote $\ell_{ij} = \ell(s_i^+ - s_j^-), p_j = \mathbb{P}(\mathbf{x}_j^-;\mathcal{D}^-), \beta = \lceil \alpha |\mathcal{D}^-| \rceil$, $\mathcal{D}^-_\beta=\mathcal{D}^-\backslash$ {$ \mathbf{x}_\{(\beta)\}$}, and $\mathcal{I}_\beta = \mathcal{D}^+ \times \mathcal{D}^-_\beta$; denote $\hat{\mathbb{E}}_\{\mathbf{x}^+ \in \mathcal{D}^+\} [z]$ as the empirical expectation of $z$ over $\mathcal{D}^+$, and similarly for $\hat{\mathbb{E}}_\{\mathbf{x}^- \in \mathcal{D}^-_\beta\} [z]$, $\hat{\mathbb{E}}_\{\mathbf{x}^+, \mathbf{x}^- \in \mathcal{I}_\beta\} [z]$; denote $\sigma = Var_\{\mathbf{x}_i^+\in\mathcal{D}^+, \mathbf{x}_j^-\in\mathcal{D}^-\}[s_i^+ - s_j^-]$, $\delta_\{ij\} = (s_i^+ - s_j^-) / \sigma$, $\bar{\delta_i} = (s_i^+ - \sum\limits_\{\mathbf{x}_j^- \in \mathcal{D}^-\}s_j^-p_j)/\sigma$.
> Assume $\sum\limits_\{\mathbf{x}_j^- \in \mathcal{D}^-\} p_j = 1$. We have $\hat{\mathcal{R}^\ell_p} \rightarrow \hat{\mathcal{R}^\ell}$ when $\sigma \rightarrow 0$ if the following conditions are met:*
>
> &emsp;&emsp; *(a) $\ell$ is convex, monotonically decreasing, $\ell(0) > 0$, and $(\ell')^2 - \ell''\cdot \ell \geq 0$;*
>
> &emsp;&emsp; *(b) $\sum\limits_{\mathbf{x}_j^- \in \mathcal{D}^-}s_j^-p_j \leq s_\beta^- $;*
>
> &emsp;&emsp; *(c) $\inf_\{u\in(0,1),v=1-u\}[A_u - B_v] \leq 0$, where*
>
> &emsp;&emsp;&emsp; *$A_u = ({|\mathcal{D}^-| - 1})(\hat{\mathbb{E}}_{\mathbf{x}^+,\mathbf{x}^- \in \mathcal{I}_\beta}[\ell^{1/u}])^{u},$*
>
> &emsp;&emsp;&emsp; *$B_v = (1 - p_\beta) \hat{\mathbb{E}}_\{\mathbf{x}^+ \in \mathcal{D}^+\}[\ell_\{i\beta\}] / (\hat{\mathbb{E}}_\{\mathbf{x}^- \in \mathcal{D}^-_\beta\}[p^{1/v}])^{v}.$*
>
> Thm. 1 shows that $\hat{\mathcal{R}^\ell_p}$ could be an effective estimation of $\hat{\mathcal{R}^\ell}$, especially when the score function $f$ converges. Thus, minimizing Eq. (6) could also minimize Eq. (5) when $\sigma$ goes to zero, i.e., both the positive examples and negative examples have zero variance.
>
> Next, we discuss the three hypotheses. We choose logistic function as our surrogate function, i.e., $\ell(x) = \log(1 + e^{-x})$, which meets the condition *(a)*: $(\ell^2) - \ell''\ell = \left(1 - e^x\cdot\log(1 + e^{-x})\right) / (1 + e^x)^2 \geq 0$. What's more, an intuitive explanation of condition *(b)* is that the weighted average of all negative scores is less than the $\beta$-largest score, which holds considering $\beta$ is often small.
>
> To verify  condition *(c)*, we also provide an experiment on synthetic data. Specifically, the scores are sampled from Gaussian distribution: {$s_i^+$ } $_i \stackrel{i.i.d}{\sim}\mathcal{N}(0.7, \sigma)$, \{$s_i^-$\}$_i \stackrel{i.i.d}{\sim}\mathcal{N}(0.3, \sigma)$, and we show in Fig. 4 how $\hat{\mathcal{R}^\ell}$ and $\hat{\mathcal{R}^\ell_p}$ change with $\sigma$ decreasing. It could be told that $\hat{\mathcal{R}^\ell}$ and $\hat{\mathcal{R}^\ell_p}$ tend to be consistent with $\sigma$ goes to zero.
>
> Link to Fig. 4: <https://ibb.co/pQSPJZc>
>
> &nbsp;
>
> ### **Q2**:
> How much does each level of approximation improve the performance/runtime from (5) to (6), and cross-batch cache?
> ### **A2**:
> One of the main reasons for proposing Eq. (6) is that optimizing Eq. (5) in a mini-batch manner is infeasible. We tested the time consumption of selecting negative examples in a full-batch manner: about 173 min per epoch, while the approximated version only needs 5 min per epoch. Moreover, even if we introduce the caching mechanism to speed up negative sample selection, the training processing still fails to converge if we use Eq. (5) directly since it ignores most of the negative examples.
>
> &nbsp;
>
> ### **Q3**:
> The runtime comparison to baselines.
> ### **A3**:
> The inference time of all methods is the same due to the same network architecture. Moreover, under the default setting, we list the training time of those methods as follows:
>
> --------------------------------training-time-for-an-iteration-(ms)---------------------------------
>
> |CE|NNRank|F&H|DDAG|SoftLabel|F&H+AUC|Eban’s|ours|
> |:---:|:---:|:---:|:---:|:---:|:---:|:---:|:---:|
> |74|74|74|81|74|74|78|84|
>
> The proposed method requires a slightly longer training time, but the gap is not obvious.
>
> &nbsp;
>
> ### **Q4**:
> Other ways to estimate the probability weighting improving the performance in practice, beyond the random walk hypothesis made.
> ### **A4**:
> As shown in Thm. 1, the probability weighting strategy is preferred as long as the three conditions are met. Despite the implementation of weighting could be varied, the proposed weighting based on the random walk hypothesis is observed from experiments and is supported by hypothesis test (please refer to Appendix B).

---

### Official Review · Reviewer_H8zB · 2021-07-15

**Rating:** 8
**Confidence:** 4

**Summary:**

In many practical applications such as medical diagnosis, the machine learning models are often required to has a low FPR. Motivated by this, the paper proposes a novel framework to optimize the true positive rate at a fixed false positive rate (TPR@FPR) for multipartite ranking. In their framework, the author proposes a surrogate optimization problem and a rank estimation technique to directly optimize TPR@FPR.

**Limitations And Societal Impact:**

Yes

**Main Review:**

1.This paper is well-written with formal derivation and the overall motivation is clear.

2. I appreciate that the optimization for TPR@FPR is challenging, especially when one has to use it to train a deep model in an end-to-end manner.

3. It’s interesting to see that the proposed framework overcomes this with a series of approximation schemes. Moreover, the observation that the short time dependence on historical updates can be reduced to white noises also helps to establish a simple yet effective probability model.

4. I find some minor issues in the paper:
  - In line 81, “multilayer perceptron (MAP)” should be “multilayer perceptron (MLP)”;
  - In line 84, the verb is missing in “Other early literature SVM”.

Overall, this paper presents a solid work, which I really enjoy reading. I recommend accepting this paper.

Questions:
- There is a significant gap between the experimental results on DR and DR-flipped, even with the same \alpha. What’s the main difference between these two versions?
- What is the specific meaning of “practical meaning of this dataset” in line 259?


**Time Spent Reviewing:**

6 hours

---

> ### Author Response · Authors · 2021-08-10
> **Response to Reviewer H8zB**
>
> We appreciate your effort and positive comments. We provide clarification on the following issues:
>
> ### **Q1**:
> What is the main difference between DR and DR-flipped?
> ### **A1**:
> These two datasets are the same except for the order of categories. For DR and DR-flipped, the categories are {0-No DR, 1-Mild DR, 2-Moderate DR, 3-Severe DR, and 4-Proliferative DR}, and {0-Proliferative DR, 1-Severe DR, 2-Moderate DR, 3-Mild DR, and 4-No DR}, respectively. TPR@FPR in DR means how many patients we can find when trying not to diagnose the normal cases as diseased, while DR-flipped is the opposite. Since the dataset is imbalanced (there are fewer cases with DR than normal cases), DR-flipped is more challenging.
>
> &nbsp;
>
> ### **Q2**:
> What is the specific meaning of “practical meaning of this dataset” in line 259?
> ### **A2**:
> Under the scenario of medical diagnosis, machine learning models are usually used for preliminary screening, so it is expected that the true positive rate can be increased as much as possible while the false negative rate keeps small. The definition of "positive" in DR-flipped is consistent with this situation.
>
> &nbsp;
>
> ### **Q3**:
> Some typos in line 81 and line 84.
> ### **A3**:
> Thanks for pointing out these typos. In line 84, "Other early literature SVM.." -> "Other early literature modifies SVM..".

---

> > ### Comment · Reviewer_H8zB · 2021-08-22
> > **My comments have been addressed.**
> >
> > After reading the authors' responses, my main concern about the difference between DR and DR-flippded has been addressed. Furthermore, I read the other reviewers' comments. And I still think the paper has merit. Therefore, I tend to keep my initial score of this paper, which can be accepted.

---

### Official Review · Reviewer_spF1 · 2021-07-16

**Rating:** 7
**Confidence:** 5

**Summary:**

The authors propose to optimize an interesting metric named TPR@FPR in multipartite ranking. This work approximates the original constraint on FPR to a probability-based loss function and speeds up the ranking estimation with embedding caches. With the above techniques, it is available to optimize TPR@FPR for deep models in an end-to-end manner.

**Limitations And Societal Impact:**

Yes

**Main Review:**

The main strengths of this paper are:
1) TPR@FPR is a useful metric in practical applications like biometrics, as FPR < \alpha mirrors the security requirements of those systems.
2) It is non-trivial to derive a suitable objective function to optimize TPR@FPR, since the quantile constraints are tricky for SGD. I like in particular how the proposed method avoids this difficult problem by proposing an arguably simple learning objective, which makes it possible to optimize TPR@FPR in deep models.
3) The effectiveness of the method is verified by experiments in different applications, and the experiment detail is clear. The sensitivity analysis helps to select hyperparameters.
4) The limitations are clearly discussed.

My two major concerns with the paper are:
1) It is mentioned in line 280 that the hyperparameters \sigma and Nc are tuned on the validation set, do other baselines have access to the validation set? If not, the comparison might be unfair.
2) The proposed method requires more memory to store the embeddings, could you please provide the analysis on space cost? Although the time complexity is present in line 245, it would be clearer if the exact time consuming is shown.

Other minor issues:
1) Equ.(8) is used to estimate ranking on a mini-batch (in line 230), but in my understanding, it should be on the whole training set.
2) Equ.(9) uses the inner product to measure the similarity, are the embeddings are normalized? It’s unreasonable if not.
3) A typo in line 280: “with learning increasing..” -> “with learning rate increasing..”.


**Time Spent Reviewing:**

5

---

> ### Author Response · Authors · 2021-08-10
> **Response to Reviewer spF1**
>
> Thank you for your positive comments, we would like to make the following response:
>
> ### **Q1**:
> Do other baselines have access to the validation set?
> ### **A1**:
> We follow the convention that the validation set is only used to tune the hyperparameters. The hyperparameters of other baselines are also tuned with the validation set, including some unique to the baselines, e.g., the learning rate of Lagrange multiplier in Eban’s. Therefore, it is fair to compare with baselines.
>
> &nbsp;
>
> ### **Q2**:
> Memory cost to store embeddings.
> ### **A2**:
> Take the DR dataset as an example, we set $N_c = 512$, $M=5$, and the embedding dimension is 2560, it takes a negligible memory of 5 MB to store these embeddings.
>
> &nbsp;
>
> ### **Q3**:
> What is the exact time-consuming of the caching mechanism?
> ### **A3**:
> In our default setting, the time spent on feature extraction and the last FC layer (which maps embeddings to scores) are 67 ms and 0.3 ms, respectively. Therefore, the caching mechanism, which saves the time of embedding recalculation, is much faster. If we remove the cache and update embeddings online instead, it will take about 3,400 (67 x 512(feature dim per class)*5 (No. of Classes) /64 (batch size))ms per iteration, while the caching version only needs 84 ms per iteration.
>
> &nbsp;
>
> ### **Q4**:
> Eq. (8) should be on the whole training set.
> ### **A4**:
> Thanks for pointing out this issue. We made corrected the typo in line 230: "$\mathbf{x}_i^{k-} \in \mathcal{B} \cap \mathcal{D}^-_k$" -> "$\mathbf{x}_i^{k-} \in \mathcal{D}^-_k$."
>
> &nbsp;
>
> ### **Q5** ###
> Use the inner product to measure similarity in Eq. (9).
> ### **A5** ###
> The embeddings are L2 normalized, thus the inner product and cosine similarity are equivalent. We will clarify this in the latest version.

---

> > ### Comment · Reviewer_spF1 · 2021-09-01
> > **Keep my rating**
> >
> > Thanks to the authors for providing a detailed response. Most of my concerns are now addressed. After reading the responses of other reviewers, I found the proposed framework could improve TPR@FPR without dropping the efficiency or the performance of traditional metrics. More interestingly, the authors also present a stronger theoretical result toward the approximation. Therefore, I keep my initial recommendation for acceptance. I strongly suggest the authors include the extra results and theoretical analysis in an updated version.

---

### Official Review · Reviewer_e74h · 2021-08-01

**Rating:** 7
**Confidence:** 4

**Summary:**

In this paper, the authors aim to optimize the True Positive Rate (TPR) at a fixed False Positive Rate (FPR) for multipartite ranking, denoted as TPR@FPR. They considered a new evaluation metric TPR@FPR for multipartite ranking in FPR sensitive scenarios. This metric focuses on model performance with a low FPR, which is consistent with practical requirements. To optimize this metric, they propose cross-bnatch approximation for multipartite ranking.

**Ethics Review Area:**

["I don’t know"]

**Limitations And Societal Impact:**

Yes

**Main Review:**

In this paper, the authors aim to optimize the True Positive Rate (TPR) at a fixed False Positive Rate (FPR) for multipartite ranking, denoted as TPR@FPR. They considered a new evaluation metric TPR@FPR for multipartite ranking in FPR sensitive scenarios. This metric focuses on model performance with a low FPR, which is consistent with practical requirements. To optimize this metric, they propose cross-batch approximation for multipartite ranking. They have conducted extensive experiments on three real-world benchmarks to show the effectiveness of the proposed method. I have the following concerns.
(1) The proposed algorithm can achieve more improvement under false positive rate sensitive application scenarios. What if we apply the proposed algorithm to other scenarios? If the data is black to the end-users, how to determine what algorithm to use?
(2) The three datasets used in this paper are small-scale. Is it possible to apply the proposed algorithm to large-scale dataset?
(3) How the feature extractor is used in this paper is not clear. Did the authors fix all the layers of the feature extractor, or did the authors tune some layers of the feature extractor?
(4) The proposed algorithm focused on improving TPR@FPR. The reviewer is very interested to see what the performance will be if we evaluate in terms of traditional evaluation metrics.

**Time Spent Reviewing:**

6

---

> ### Author Response · Authors · 2021-08-10
> **Response to Reviewer e74h**
>
> We appreciate your time and detailed comments on our manuscript. We would like to reply to the following questions:
>
> ### **Q1**:
> How to determine what algorithm to use if the data is black to the end-users?
> ### **A1**:
> The choice of algorithm depends on the metric to be used when making a decision. It might occur that the data is black, and we cannot access them explicitly. In this case, the data releaser should at least either provide the metric to be optimized or provide sufficient prior knowledge, say, whether the classes are imbalanced, or whether there are sufficient hard examples. If one knows that the metric to be optimized is TPR@FPR (a common choice for app, such as Medical Analysis, Face recognition, Fraud Detection), or the dataset is imbalanced with a large amount of hard examples, our algorithm can then come into play.
>
> &nbsp;
>
> ### **Q2**:
> Application to large-scale datasets.
> ### **A2**:
> Benefiting from the large capacity of deep models, the proposed algorithm can be applied to large-scale datasets. We have conduct extra experiments on a facial age estimation dataset with 217k images (<https://www.kaggle.com/mariafrenti/age-prediction>), and the results are as follows:
>
> ----------TPR@FPR-on-AgePrediction-(%)----------
>
> |FPR|0.003|0.01|0.03|
> |:--:|:--:|:--:|:--:|
> |CE|11.3|38.6|53.1|
> |F&H|11.0|36.4|48.9 |
> |SoftLabel|11.0|37.5|52.1|
> |F&H+AUC|12.4|39.6|53.8|
> |Eban’s|11.7|39.3|54.3|
> |Ours|**16.0**|**40.2**|**56.2**|
>
> As shown in the table, our method remains effective in this dataset compared to others, especially when the expected FPR is low.
>
> &nbsp;
>
> ### **Q3**:
> How to use the feature extractor?
> ### **A3**:
> In our experiments, we tune all the parameters of the feature extractor with backpropagation.
>
> &nbsp;
>
> ### **Q4**:
> What is the performance if we evaluate in terms of traditional evaluation metrics?
> ### **A4**:
> Here we report the commonly-used AUC metric:
>
> ----------AUC-on-three-benchmarks-(%)----------
>
> ||DR|UTKFace|NSFW|
> |:--:|:--:|:--:|:--:|
> |CE|84.4|94.6|96.6|
> |NNRank|85.4|94.5|97.5|
> |F&H|85.8|95.7|97.4|
> |CR-DDAG|84.2|93.7|96.1|
> |PR-DDAG|85.0|96.0|97.1|
> |SoftLabel|85.9|**96.0**|97.6|
> |F&H+AUC|85.6|95.6|97.3|
> |Eban’s|85.4|95.4|97.7|
> |Ours|**86.3**|95.9|**98.0**|
>
> Compared with the competitors, the proposed method achieves comparable performance on AUC. This shows that our method can improve the processing capacity of hard examples without sacrificing overall performance.

---

### Decision · Program_Chairs · 2021-09-27

**Decision:**

Accept (Poster)

**Comment:**

TPR@FPR is a useful metric in many important practical applications (e.g. biometrics). The paper made a valuable contribution on the challenging problem of optimizing TPR@FPR for multipartite ranking.   On the top of deep learning models,  it proposed a novel framework by introducing a differentiable surrogate and a fast ranking estimation method to optimize the non-decomposable objective function. Overall, the problem is well motivated and the proposed framework/algorithms for optimizing TPR@FPR are novel and interesting.